# Oligopaint DNA FISH reveals telomere-based meiotic pairing dynamics in the silkworm, *Bombyx mori*

**Leah F. Rosin**[1]*, **Jose Gil, Jr.**[2], **Ines A. Drinnenberg**[2], **Elissa P. Lei**[1]*

**1** Nuclear Organization and Gene Expression Section; Laboratory of Biochemistry and Genetics, National Institute of Diabetes and Digestive and Kidney Diseases, National Institutes of Health, Bethesda, Maryland, United States of America, **2** Institut Curie, PSL Research University, CNRS, Paris, France; Sorbonne Université, Institut Curie, CNRS, Paris, France

* leah.rosin@nih.gov (LFR); leielissa@niddk.nih.gov (EPL)

**Data Availability Statement:** All relevant data are within the manuscript and its Supporting Information files.

**Funding:** This work was funded by the Intramural Program of the National Institute of Diabetes and

## Abstract

Accurate chromosome segregation during meiosis is essential for reproductive success. Yet, many fundamental aspects of meiosis remain unclear, including the mechanisms regulating homolog pairing across species. This gap is partially due to our inability to visualize individual chromosomes during meiosis. Here, we employ Oligopaint FISH to investigate homolog pairing and compaction of meiotic chromosomes and resurrect a classical model system, the silkworm *Bombyx mori*. Our Oligopaint design combines multiplexed barcoding with secondary oligo labeling for high flexibility and low cost. These studies illustrate that Oligopaints are highly specific in whole-mount gonads and on meiotic squashes. We show that meiotic pairing is robust in both males and females and that pairing can occur through numerous partially paired intermediate structures. We also show that pairing in male meiosis occurs asynchronously and seemingly in a transcription-biased manner. Further, we reveal that meiotic bivalent formation in *B. mori* males is highly similar to bivalent formation in *C. elegans*, with both of these pathways ultimately resulting in the pairing of chromosome ends with non-paired ends facing the spindle pole. Additionally, microtubule recruitment in both *C. elegans* and *B. mori* is likely dependent on kinetochore proteins but independent of the centromere-specifying histone CENP-A. Finally, using super-resolution microscopy in the female germline, we show that homologous chromosomes remain associated at telomere domains in the absence of chiasma and after breakdown and modification to the synaptonemal complex in pachytene. These studies reveal novel insights into mechanisms of meiotic homolog pairing both with or without recombination.

## Author summary

Meiosis is the specialized cell division occurring exclusively in ovaries and testes to produce egg and sperm cells, respectively. The accurate distribution of chromosomes (the genetic material) during this process is essential to prevent infertility/sterility and developmental disorders in offspring. As researchers are specifically unable to study the

Digestive and Kidney Diseases, National Institutes of Health (NIDDK; DK015602 to E.P.L.), the Eunice Kennedy Shriver National Institute of Child Health and Human Development, National Institutes of Health (NICHD; 1K99HD104851 to L.F.R), LabEx DEEP grants from the Agence National de la Recherche (ANR-11-LABX-0044 and ANR-10-IDEX-0001-02 to I.A.D.), an ATIP-AVENIR research grant from CNRS/INSERM, the European Research Council (ERC; CENEVO-758757 to I.A.D), and Institut Curie (I.A.D). NIDDK provides salary support for E.P.L. and L.F.R. Institute Curie and CNRS provide salary support for I.A.D. ERC provides salary support for J.G. The funders had no role in study design, data collection and analysis, decision to publish, or preparation of the manuscript.

**Competing interests:** The authors have declared that no competing interests exist.

mechanisms regulating meiosis in depth in humans, identifying broadly conserved aspects of meiotic chromosome segregation is essential for making accurate inferences about human biology. Here, we use a sophisticated chromosome painting approach called Oligopaints to visualize and study chromosomes during meiosis in the silkworm, *Bombyx mori*. Using this method, we show that chromosome ends (telomeres) play a central role in regulating interactions between maternal and paternal chromosome copies during both sperm and egg development.

## Introduction

Precise homolog pairing and unpairing during meiosis is essential for genetic recombination and accurate chromosome segregation. Errors in chromosome segregation during meiosis can lead to reduced fertility, miscarriages, or chromosomal disorders in progeny, such as Down or Turner Syndromes [1]. Decades of research have gone into characterizing the synaptonemal complex (SC), a ubiquitous, proteinaceous structure that holds homologs together during meiotic prophase [2, 3]. Yet how homologs find each other and come together in three dimensional (3D) space is still poorly understood. One of the main reasons that homolog pairing has remained such an enigma is the lack of cytological tools available for assaying chromosome- and locus-specific pairing dynamics during meiosis. Several recent studies have taken advantage of advances in super-resolution microscopy techniques, such as Structure Illumination Microscopy (SIM) and Stochastic Optical Reconstruction Microscopy (STORM), to visualize meiotic pairing in more detail than ever before [4–9]. However, these approaches have been limited to studying pairing genome-wide by fluorescently labeling elements of the SC [5, 7–12] or to visualizing small genomic loci by FISH [13–16].

Recent technological innovations in the design and synthesis of specialized DNA FISH probes called Oligopaints have made visualizing whole, individual chromosomes or complex sub-chromosomal loci in meiotic cells feasible. Unlike traditional BAC-based FISH probes, Oligopaints are computationally designed based on genome sequence data [17, 18]. This approach allows for only unique, single copy sequences to be labeled, significantly increasing the specificity and resolution of FISH. Here, we leverage the flexibility of the Oligopaint design to add barcodes to label either whole chromosomes or different sub-chromosomal loci using the same set of oligos, as previously described [19]. This multiplexed approach allows for different highly specific FISH probes to be generated at low cost and high throughput. Oligopaints and related oligo-based FISH approaches have previously been used for karyotype analyses or characterization of interphase chromosome dynamics in *Drosophila*, *C. elegans*, mammals, and plants [16, 19–31]. Recently, similar approaches have also been applied to the study of small chromosomal loci during meiosis [32, 33], but Oligopaints have never before been used to characterize compaction and pairing of multiple, whole chromosomes during meiosis. Finally, this is the first study to use Oligopaints to visualize chromosomes in Lepidoptera (moths and butterflies).

Here, we combine Oligopaint DNA FISH with one of the first model systems used to study meiotic chromosomes, the silkworm moth. *Bombyx mori* are holocentric insects, with centromeres that form all along the chromosome during mitosis [34–37]. The holocentric mitotic configuration is also seen in many plants and nematodes, including *C. elegans* [38–40]. However, the holocentric chromosome configuration prevents accurate biorientation of bivalents formed after recombination and is therefore incompatible with canonical meiosis [40, 41]. Instead, chromosomes in holocentric organisms often display "telokinetic" or

"telokinetic-like" chromosomes during meiosis, where kinetochore activity is restricted toward telomere domains [40, 42–47]. In *C. elegans*, crossover position dictates which telomere faces poleward to connect to the spindle microtubules [42, 45, 48–50]. A similar telokinetic mechanism for segregating meiotic chromosomes was also previously hypothesized to occur in *B. mori* [51–53] but has never before been directly observed. Furthermore, meiotic segregation in *C. elegans* occurs in the absence of the centromere-specifying factor Centromere Protein A (CENP-A) [50]. Instead, microtubules either run parallel to chromosomes to facilitate segregation or connect directly to cup-shaped kinetochores that form at chromosome ends [46, 54]. Interestingly, CENP-A is entirely absent from the genomes of butterflies and moths [55]. How moths and butterflies segregate chromosomes during meiosis in the absence of CENP-A remains to be explored.

Unlike *B. mori* spermatogenesis, which has been reported to support crossovers and canonical pairing, oogenesis in *B. mori* is unconventional. Chiasmata are not observed in female meiosis in silkworms. Furthermore, the central elements of the SC break down just after pachytene (one of the sub-stages of meiotic prophase I), and the lateral elements of the SC are thought to be completely remodeled to form masses of protein (modified SC) between the two homologs [53, 56–58]. This modified SC is reported to be greater than one micron in width by the end of prophase I, thereby ultimately undoing tight end-to-end homolog pairing while still holding homologs together until anaphase I [58]. Thus, pairing along the entire length of the chromosomes is not expected after pachytene.

Our studies here illustrate that Oligopaints are robust and specific in germline cells and can be used to visualize chromosomes even in unconventional model systems with draft genomes. Using this FISH-based approach, we demonstrate that pairing in early meiotic prophase in the *B. mori* male germline is initiated at chromosome ends, with gene-rich chromosome ends more often initiating pairing. We further show that telomeric regions face poleward during metaphase I and act as localized kinetochores during male meiosis, and both telomeres on any given chromosome harbor the ability to act as local kinetochores. Additionally, using immunofluorescence combined with FISH, we reveal that the DNA-binding kinetochore protein Centromere Protein T (CENP-T) is present on meiotic chromosomes and likely facilitates microtubule attachments at metaphase I. Finally, using super-resolution microscopy combined with Oligopaints in the female germline, we show that homologs remain paired at telomere regions throughout meiotic prophase I even after remodeling of the SC. Overall, we provide the first extensive characterization of whole and sub-chromosome dynamics during meiosis in any species, thereby pioneering the use of Oligopaints as a tool for studying meiotic pairing and progression.

## Results

### *B. mori* Oligopaint design

To visualize chromosomes in the silkworm *B. mori*, we designed and generated Oligopaint libraries targeting six of the 27 autosomes and the Z sex chromosome. We chose chromosomes of varying size and gene density (Table 1) so we could explore whether these properties affect the dynamics of meiotic chromosomes. We were also interested in examining the sex chromosomes to determine whether they behave differently than autosomes in meiosis. Our Oligopaint libraries were designed using the Oligominer pipeline [18, 59] based on the updated 2019 silkworm genome assembly [60]. Oligos were designed with 80 bp of homology and map to the genome only once (therefore only labeling unique, single copy sequences). As the W sex chromosome is largely composed of repetitive sequences, it was not suitable for the Oligopaint design strategy utilized here, and therefore we focused on autosomes and the Z chromosome.

**Table 1. Chromosome and chromosome paint information.**

| Chrom | chromosome size (bp) | size painted (bp) | paint start | paint stop | density (probes/kb) | total # of oligos | gene density (genes per Mb) |
|---|---|---|---|---|---|---|---|
| 4 | 18737234 | 18639239 | 282 | 18639521 | 1.5 | 26841 | 39.7 |
| 7 | 13944894 | 13868845 | 35931 | 13904776 | 3 | 42625 | 36.0 |
| 15 | 18440292 | 18354755 | 21089 | 18375844 | 1 | 17756 | 43.6 |
| 16 | 14337292 | 14275583 | 27737 | 14303320 | 1.5 | 20190 | 43.7 |
| 17 | 16840672 | 16806551 | 9415 | 16815966 | 1.5 | 23834 | 38.3 |
| 23 | 21465692 | 21339065 | 123188 | 21462253 | 1.5 | 30506 | 35.8 |
| Z/1 | 20666287 | 20578020 | 37936 | 20615956 | 1.5 | 29784 | 32.2 |

This Oligopaint design approach yielded a maximum probe density of approximately 3 oligos per kilobase (kb) of DNA. For most chromosomes in our analysis, we further reduced this density to 1 or 1.5 probes per kb (Table 1), a probe density that has been shown previously to be sufficient for whole chromosome paints [30]. The resulting oligos are fairly evenly distributed along each chromosome, with gaps in regions where repetitive sequences are more abundant (Fig 1A–1C). These oligo libraries were then multiplexed as described previously [19] with one or more barcode sequences to allow for the amplification of individual chromosomes, sub-chromosomal stripes, and/or active and inactive chromatin domains (Tables 1–5 and Figs 1B, 1C, and S1). In total, the libraries consisted of 191,536 oligos (designated as "primary oligos") up to 160 bp in length, which includes the 80 bp of homology, up to two unique 20 bp sub-chromosomal barcodes, and two 20 bp whole chromosome universal barcodes (Fig 1D). During the PCR amplification steps, secondary oligo binding sites are added to the primary oligos, to which fluorescently labeled secondary oligos anneal during the FISH protocol (Fig 1E; [21, 30]). This method allows for increased flexibility when combining probes for multi-channel imaging. Together, this multiplexed probe design combined with secondary oligo labeling both increases the efficiency of Oligopaint labeling and reduces the cost of Oligopaint synthesis.

## *B. mori* Oligopaints are highly specific

As the *B. mori* genome used to design the Oligopaints is in a semi-draft state (assembled into chromosomes but with many unmapped contigs), we first tested the specificity of our *B. mori* chromosome paints using karyotype analysis. Due to the small, holocentric nature of silkworm chromosomes, mitotic chromosomes are highly compact, while chromosomes in meiotic prophase I (pachytene sub-stage; Fig 2A; reviewed in [61]) are more linear due to synapsis (S2 Fig)[62]. Therefore, meiotic chromosomes are better suited for karyotype analyses. As silkmoths have a very short adult lifespan (only 5–7 days), meiosis begins early in the larval stages [63]. Hence, meiotic chromosomes from late 4[th] or early 5[th] instar larval testes and ovaries were visualized using a squashing technique. This approach uses a hypotonic shock combined with physical squashing to burst cells and disperse nuclei onto the slide, retaining only a minimal amount of spatial and temporal information from the tissue. A detailed description of our meiotic squash protocol can be found in the Materials and Methods. Since homologs are paired during most sub-stages of meiotic prophase I (Fig 2A), we expected a single fluorescence signal per chromosome for a given probe.

Using our whole chromosome paints, three chromosomes at a time were labeled on meiotic squashes from testes and ovaries. We were able to easily identify cells in the pachytene stage, when homologs are tightly paired end-to-end, using a combination of DAPI structure and chromosome morphology. Indeed, we observed singular and distinct fluorescence signals for

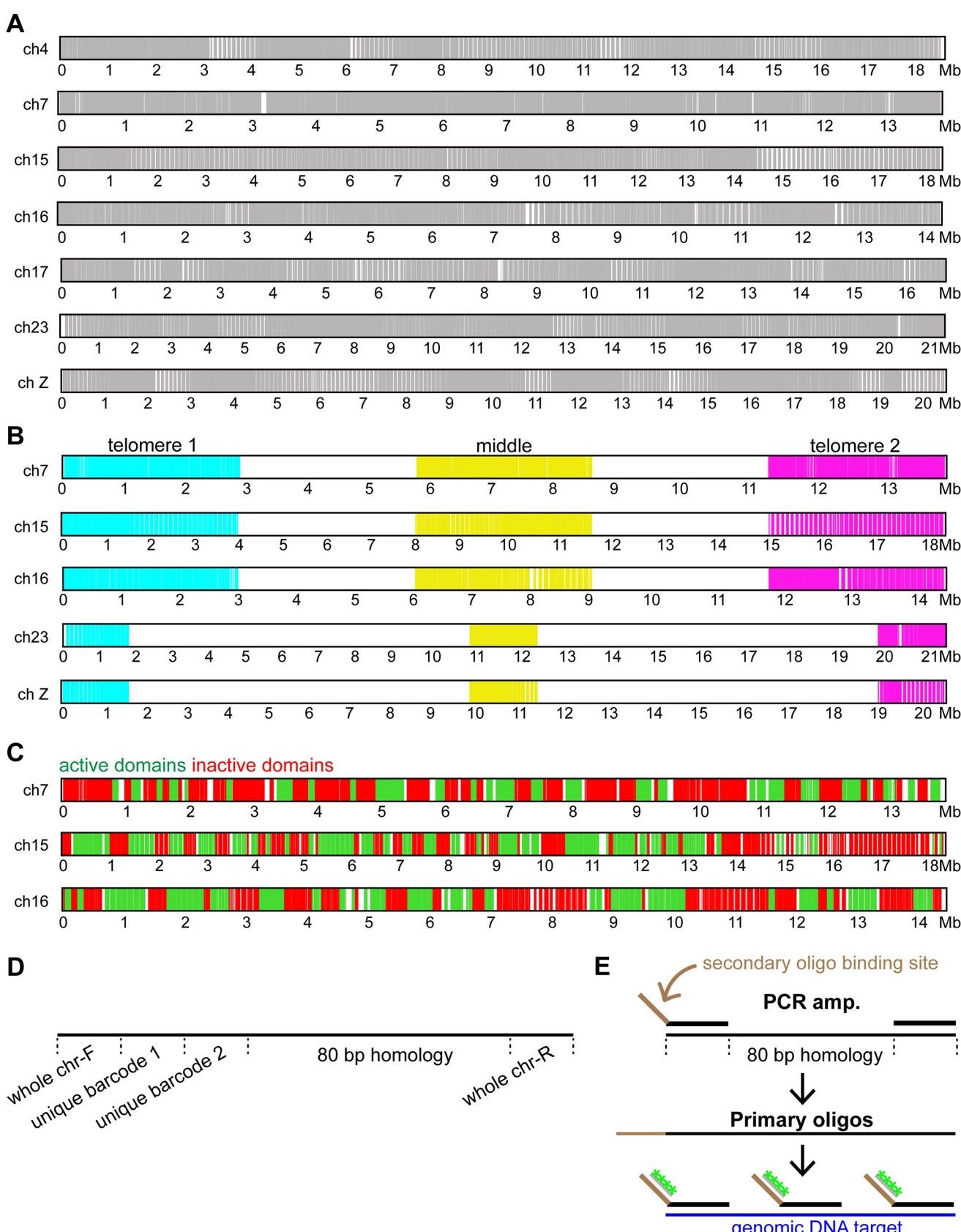

**Fig 1. *B. mori* Oligopaint design. (A-C)** Schematic of Oligopaints in *B. mori*. Whole chromosome Oligopaints are shown in A, stripe Oligopaints in B, and active/inactive Oligopaints in C. White regions indicate the absence of oligos (A) or regions not labeled by the respective barcode indices (B, C). **(D)** Schematic of primary probe design, showing whole chromosome barcodes and two unique barcodes (for stripes or active/inactive domains). **(E)** Schematic for Oligopaint DNA FISH assay with labeled secondary oligos. First, ordered oligos are amplified with primers containing barcode of interest and secondary oligo binding site, generating primary oligos. Primary oligos are then annealed to DNA and labeled with secondary oligos (shown in green).

each tested chromosome at pachytene, where DNA is in a loose lampbrush configuration (Fig 2B–2E and 2G). In our larval testes squashes, we were also able to identify cells earlier in meiotic prophase I before pairing has completely occurred, in which the DAPI stain shows a dense lampbrush structure (leptotene/zygotene; Fig 2F), as well as cells in late prophase (diplotene/diakinesis; Fig 2H) in which chromosomes are more condensed, paired, and have diffuse DAPI signal. Of cells in Prophase I in late 4th-early 5th instar larval testes, the majority are in pachytene (Fig 2I). Moreover, we were able to identify cells in which chromosomal bivalents are compacted and aligned at the metaphase plate (metaphase I, Fig 2J). Finally, in addition to meiotic cells, we observed interphase and mitotic cells in our testes squashes harboring two distinct, round fluorescence signals per chromosome, indicating that homologs are unpaired in the majority of non-meiotic cells in *B. mori* (Figs 2K, S3, and S4).

In larval ovary squashes, we were also able to identify linear, paired pachytene chromosomes (Fig 2E). At this early pachytene stage, both oocyte- and nurse cell-destined germ cells pair their homologs. Subsequently during differentiation, nurse cell chromosomes begin to unpair. Post-pachytene nurse cells could be identified in our squashes by their unpaired but proximal, condensed chromosomes (S5 Fig; [57]). Interestingly, when we attempted to use these paints to characterize chromosome morphology in a *B. mori* ovary-derived cell line, BmN4, we found that our probes partially labeled multiple chromosomes, suggesting that the karyotype in these cells has undergone dramatic rearrangements and possible ploidy changes compared to the genome-derived strain (S6 Fig). This observation further validated the specificity of our probes and their ability to detect translocations. Together, these data not only illustrate that our Oligopaint libraries are specific but also validate the *B. mori* genome assembly.

**Table 2. Stripe sub-library paint information.**

| Chrom. and stripe | paint start | paint stop | Stripe size (Mb) |
| --- | --- | --- | --- |
| 7 tel1 | 35931 | 2809682 | 2.77 |
| 7 mid | 5587874 | 8357288 | 2.76 |
| 7 tel2 | 11131025 | 13904776 | 2.77 |
| 15 tel1 | 21089 | 3684775 | 3.66 |
| 15 mid | 7380341 | 11061106 | 3.68 |
| 15 tel2 | 14748412 | 18375844 | 3.62 |
| 16 tel1 | 27737 | 2865990 | 2.84 |
| 16 mid | 5732698 | 8598132 | 2.87 |
| 16 tel2 | 11464265 | 14303320 | 2.84 |
| 23 tel1 | 123188 | 1638259 | 1.51 |
| 23 mid | 9909172 | 11556867 | 1.65 |
| 23 tel2 | 19815237 | 21462253 | 1.65 |
| Z tel1 | 37936 | 1589202 | 1.55 |
| Z mid | 9538407 | 11119201 | 1.58 |
| Z tel2 | 19078122 | 20615956 | 1.53 |

**Table 3. Ch7 Active and inactive chromatin domains paint information.**

| Domain | Start | Stop | Size (bp) |
| --- | --- | --- | --- |
| inactive | 35931 | 800000 | 764070 |
| active | 800001 | 900000 | 100000 |
| inactive | 1000001 | 1100000 | 100000 |
| active | 1100001 | 1250000 | 150000 |
| inactive | 1300001 | 1500000 | 200000 |
| active | 1500001 | 1600000 | 100000 |
| inactive | 1600001 | 1700000 | 100000 |
| active | 1700001 | 1850000 | 150000 |
| inactive | 1850001 | 1900000 | 50000 |
| active | 1950001 | 2000000 | 50000 |
| inactive | 2000001 | 2300000 | 300000 |
| inactive | 2400001 | 2600000 | 200000 |
| active | 2600001 | 2700000 | 100000 |
| inactive | 2700001 | 3350000 | 650000 |
| active | 3400001 | 3650000 | 250000 |
| inactive | 3650001 | 3900000 | 250000 |
| active | 3900001 | 4000000 | 100000 |
| inactive | 4000001 | 4550000 | 550000 |
| active | 4550001 | 4650000 | 100000 |
| inactive | 4650001 | 4950000 | 300000 |
| active | 4950001 | 5400000 | 450000 |
| inactive | 5450001 | 5800000 | 350000 |
| active | 5900001 | 6050000 | 150000 |
| inactive | 6050001 | 6250000 | 200000 |
| inactive | 6300001 | 6450000 | 150000 |
| active | 6450001 | 6550000 | 100000 |
| inactive | 6550001 | 6650000 | 100000 |
| active | 6700001 | 6800000 | 100000 |
| active | 6850001 | 7150000 | 300000 |
| inactive | 7150001 | 7400000 | 250000 |
| active | 7400001 | 7600000 | 200000 |
| inactive | 7600001 | 7900000 | 300000 |
| active | 7950001 | 8200000 | 250000 |
| inactive | 8250001 | 8750000 | 500000 |
| inactive | 8800001 | 9050000 | 250000 |
| active | 9050001 | 9300000 | 250000 |
| inactive | 9350001 | 9400000 | 50000 |
| active | 9400001 | 9550000 | 150000 |
| inactive | 9650001 | 10800000 | 1150000 |
| active | 10850001 | 11100000 | 250000 |
| active | 11150001 | 11400000 | 250000 |
| inactive | 11400001 | 11650000 | 250000 |
| active | 11650001 | 11950000 | 300000 |
| inactive | 11950001 | 12300000 | 350000 |
| active | 12300001 | 12600000 | 300000 |
| inactive | 12600001 | 12700000 | 100000 |
| active | 12700001 | 12750000 | 50000 |

(*Continued*)

**Table 3.** (Continued)

| Domain | Start | Stop | Size (bp) |
| --- | --- | --- | --- |
| inactive | 12800001 | 12950000 | 150000 |
| active | 12950001 | 13150000 | 200000 |
| inactive | 13150001 | 13250000 | 100000 |
| active | 13350001 | 13450000 | 100000 |
| inactive | 13500001 | 13650000 | 150000 |
| active | 13700001 | 13849955 | 149955 |

**Table 4. Ch15 Active and inactive chromatin domains paint information.**

| Domain | Start | Stop | Size (bp) |
| --- | --- | --- | --- |
| inactive | 21089 | 200000 | 178912 |
| active | 250001 | 850000 | 600000 |
| active | 900001 | 1000000 | 100000 |
| inactive | 1000001 | 1400000 | 400000 |
| active | 1400001 | 1950000 | 550000 |
| inactive | 1950001 | 2250000 | 300000 |
| active | 2250001 | 2550000 | 300000 |
| inactive | 2550001 | 2800000 | 250000 |
| active | 2850001 | 3200000 | 350000 |
| inactive | 3200001 | 3500000 | 300000 |
| active | 3550001 | 3850000 | 300000 |
| inactive | 3850001 | 3900000 | 50000 |
| active | 3950001 | 4100000 | 150000 |
| inactive | 4100001 | 4250000 | 150000 |
| active | 4250001 | 4350000 | 100000 |
| inactive | 4350001 | 4650000 | 300000 |
| active | 4650001 | 4750000 | 100000 |
| inactive | 4750001 | 4950000 | 200000 |
| inactive | 5000001 | 5150000 | 150000 |
| active | 5150001 | 5250000 | 100000 |
| inactive | 5300001 | 5350000 | 50000 |
| active | 5350001 | 6100000 | 750000 |
| inactive | 6100001 | 6200000 | 100000 |
| active | 6250001 | 6500000 | 250000 |
| inactive | 6500001 | 6700000 | 200000 |
| active | 6700001 | 6850000 | 150000 |
| inactive | 6900001 | 7100000 | 200000 |
| active | 7100001 | 7300000 | 200000 |
| inactive | 7300001 | 7550000 | 250000 |
| active | 7550001 | 7800000 | 250000 |
| inactive | 7800001 | 8100000 | 300000 |
| active | 8100001 | 8300000 | 200000 |
| inactive | 8400001 | 8650000 | 250000 |
| active | 8650001 | 8750000 | 100000 |
| inactive | 8750001 | 8800000 | 50000 |
| active | 8850001 | 8900000 | 50000 |

(*Continued*)

**Table 4.** (Continued)

| Domain | Start | Stop | Size (bp) |
|---|---|---|---|
| inactive | 8900001 | 9100000 | 200000 |
| active | 9100001 | 9500000 | 400000 |
| inactive | 9500001 | 9550000 | 50000 |
| active | 9600001 | 9750000 | 150000 |
| active | 9800001 | 9950000 | 150000 |
| inactive | 10000001 | 10500000 | 500000 |
| active | 10500001 | 11200000 | 700000 |
| active | 11350001 | 11400000 | 50000 |
| inactive | 11400001 | 11500000 | 100000 |
| active | 11500001 | 11950000 | 450000 |
| inactive | 11950001 | 12000000 | 50000 |
| active | 12050001 | 12200000 | 150000 |
| inactive | 12200001 | 12250000 | 50000 |
| active | 12300001 | 12500000 | 200000 |
| inactive | 12500001 | 12600000 | 100000 |
| active | 12600001 | 12850000 | 250000 |
| inactive | 12850001 | 12950000 | 100000 |
| active | 12950001 | 13400000 | 450000 |
| inactive | 13450001 | 13600000 | 150000 |
| active | 13650001 | 13750000 | 100000 |
| inactive | 13750001 | 14150000 | 400000 |
| inactive | 14200001 | 14850000 | 650000 |
| active | 14900001 | 15050000 | 150000 |
| inactive | 15100001 | 15250000 | 150000 |
| active | 15300001 | 15500000 | 200000 |
| inactive | 15550001 | 15600000 | 50000 |
| active | 15600001 | 15750000 | 150000 |
| inactive | 15750001 | 16000000 | 250000 |
| inactive | 16050001 | 16350000 | 300000 |
| inactive | 16400001 | 17850000 | 1450000 |
| inactive | 17900001 | 18100000 | 200000 |
| active | 18150001 | 18200000 | 50000 |
| inactive | 18200001 | 18250000 | 50000 |
| active | 18300001 | 18350000 | 50000 |
| inactive | 18350001 | 18375844 | 25844 |

## Detection of stripe and chromatin state sub-libraries

While the specificity of our whole chromosome paints indicated that the *B. mori* genome is accurately assembled at the chromosome level, we needed to validate the intra-chromosomal genome assembly and in turn, the specificity of our sub-chromosomal paints. For this, we again turned to pachytene squashes in the male germline, where the linear nature of chromosomes allowed us to verify the linear order of our probes. We started with the stripe sub-libraries, wherein selected chromosomes were sub-divided into 3 Mb or 1.5 Mb domains, depending on the chromosome (Table 2 and Fig 1B). The first, middle, and last stripes were labeled with secondary oligos to visualize three stripes along each chromosome (Figs 1B and 3A–3E; telomere 1 (tel1), middle (mid), and telomere 2 (tel2), respectively). FISH with these

**Table 5. Ch16 Active and inactive chromatin domains paint information.**

| Domain | Start | Stop | Size (bp) |
|---|---|---|---|
| inactive | 27737 | 50000 | 22264 |
| active | 50001 | 150000 | 100000 |
| inactive | 150001 | 250000 | 100000 |
| active | 250001 | 350000 | 100000 |
| inactive | 350001 | 650000 | 300000 |
| active | 700001 | 1350000 | 650000 |
| inactive | 1400001 | 1700000 | 300000 |
| active | 1700001 | 2300000 | 600000 |
| inactive | 2300001 | 2400000 | 100000 |
| active | 2400001 | 2700000 | 300000 |
| inactive | 2700001 | 3200000 | 500000 |
| active | 3200001 | 3600000 | 400000 |
| inactive | 3600001 | 4050000 | 450000 |
| active | 4050001 | 4200000 | 150000 |
| inactive | 4200001 | 4500000 | 300000 |
| active | 4500001 | 4600000 | 100000 |
| active | 4700001 | 4750000 | 50000 |
| inactive | 4750001 | 4800000 | 50000 |
| active | 4900001 | 4950000 | 50000 |
| active | 5000001 | 5250000 | 250000 |
| inactive | 5250001 | 5600000 | 350000 |
| active | 5600001 | 6000000 | 400000 |
| inactive | 6000001 | 6150000 | 150000 |
| active | 6200001 | 6250000 | 50000 |
| active | 6300001 | 6500000 | 200000 |
| active | 6550001 | 6650000 | 100000 |
| inactive | 6650001 | 6850000 | 200000 |
| active | 6850001 | 7000000 | 150000 |
| inactive | 7050001 | 7950000 | 900000 |
| inactive | 8000001 | 8500000 | 500000 |
| active | 8550001 | 8750000 | 200000 |
| inactive | 8800001 | 8900000 | 100000 |
| active | 8900001 | 9400000 | 500000 |
| inactive | 9400001 | 9450000 | 50000 |
| active | 9500001 | 10100000 | 600000 |
| inactive | 10100001 | 11450000 | 1350000 |
| active | 11450001 | 11550000 | 100000 |
| inactive | 11550001 | 11900000 | 350000 |
| active | 11950001 | 12250000 | 300000 |
| inactive | 12250001 | 12400000 | 150000 |
| active | 12400001 | 12500000 | 100000 |
| inactive | 12500001 | 12700000 | 200000 |
| active | 12750001 | 13200000 | 450000 |
| inactive | 13250001 | 13800000 | 550000 |
| active | 13800001 | 14100000 | 300000 |
| inactive | 14100001 | 14303320 | 203320 |

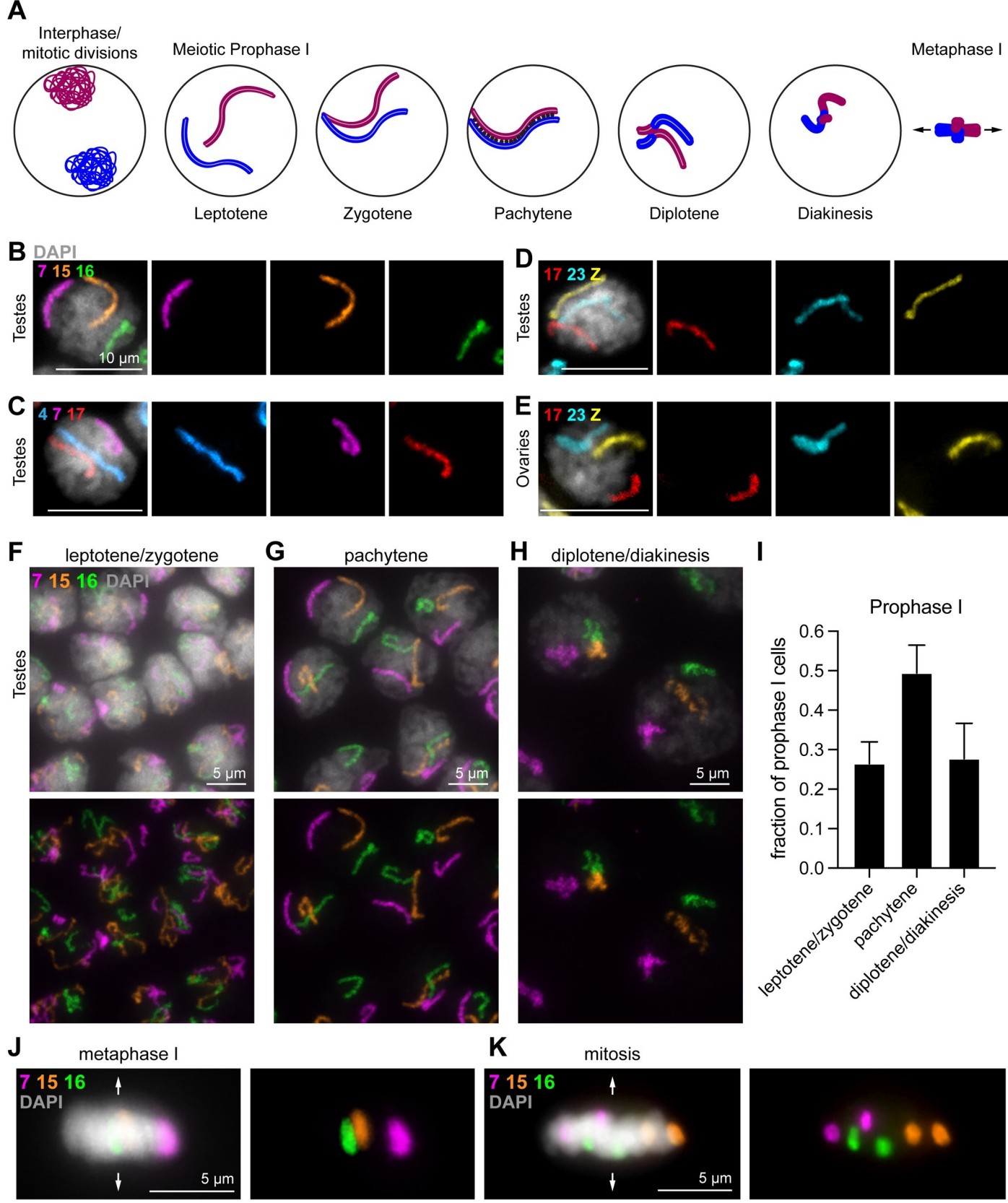

**Fig 2. Whole chromosome Oligopaints in *B. mori* 5th instar germline squashes. (A)** Schematic of early meiosis I (prophase I and metaphase I). One pair of homologous chromosomes is shown (red = paternal; blue = maternal). Prophase I is typically subdivided into five distinct stages: leptotene, zygotene, pachytene, diplotene, and diakinesis. Briefly: *in leptotene, replicated chromosomes are reorganized and compacted into a linear scaffold structure. In zygotene, synapsis begins between the homologous chromosomes. In pachytene, synapsis is complete (black dots represent the synaptonemal complex holding the homologs together). This stage is also when crossing over can occur. In diplotene, the homologs repulse, condense further, and the SC breaks down. The homologs remain attached via chiasma (crossovers). Finally, in diakinesis, chromosome condensation and cruciform bivalent formation is nearly complete as the cell prepares for metaphase I.* Arrows in metaphase I schematic indicate the assumed direction of spindle poles based on the direction of the metaphase plate. **(B-E)** Pachytene cells labeled with three whole chromosome Oligopaints, as indicated. B-D, larval testes. E, larval ovary. Scale bars = 10 μm. DAPI is shown in gray. F-H) Meiotic prophase I cells from larval testes squashes with whole chromosome paints for ch7 (magenta), ch15 (orange), and ch16 (green). Boxes indicate subsequent panels as indicated. DAPI is shown in gray. **(F)** Leptotene/zygotene cells, with unpaired, decondensed chromosomes and a dense lampbrush DNA stain. **(G)** Pachytene cells, with paired, linear, and relatively decondensed chromosomes and a loose lampbrush DNA stain. **(H)** Diplotene/diakinesis cells, with paired, less linear and more compact chromosomes and a diffuse DNA stain. **(I)** Quantification of prophase cell staging from late 4th-early 5th instar larval testes. n = 3 testes were analyzed. Error bars show mean and standard deviation between testes. **(J)** Metaphase I cells, with paired homologs condensed and aligned along the metaphase plate. Arrows indicate the direction of spindle poles. **(K)** Mitotic cells from larval testes, with chromosomes condensed and aligned along the metaphase plate but with unpaired homologs. Arrows indicate the direction of spindle poles.

stripe paints for ch15 in pachytene squashes from larval testes revealed a singular focus for each stripe, with the tel1, mid, and tel2 domains positioned in the predicted order along the linear chromosome (Fig 3A). This was also true for ch7, ch16, ch23, and chZ (Fig 3B–3E).

To test the specificity of our transcriptionally active and inactive chromatin domain paints, we labeled all three chromosomes with this barcode index (ch7, 15, and 16) at the same time on larval testes pachytene squashes. This analysis revealed three separate linear chromosomes with distinct banding patterns corresponding to the respective paint schematic (Figs 1 and 3F). Together, these results indicate that the intra-chromosomal assembly for these chromosomes is highly accurate and our paints are specific for their target chromosomal domains.

## Homolog pairing in early meiotic prophase is not synchronous

When examining the images of whole chromosome paints on meiotic squashes from testes, we noticed a wide variety of partially paired chromosome configurations (Figs 2F and 4A). We therefore attempted to quantify these configurations to determine the prevalence of each in order to ascertain whether they may be biologically relevant. We broadly classified the most common partially paired configurations as linear with loops, forked, circular, or lollipop (Fig 4A and 4B). Interestingly, the forked and circular configurations were observed most frequently for all analyzed chromosomes (Fig 4B). Furthermore, not all chromosomes within a single cell pair simultaneously, as many cells harbor some paired and some unpaired chromosomes (Fig 4C–4E). When analyzing three chromosomes at a time, less than 10% of cells show no pairing between any three examined chromosomes, while more than 40% show pairing between all three visualized homologous pairs (Fig 4D).

To determine if certain chromosomes more frequently initiate pairing earlier than others, we quantified which chromosome initiated pairing in the cells in which only one of the three labeled chromosome sets had begun to pair. This analysis revealed that large, gene-rich chromosomes (ch15, 17, and 4; Fig 4E) more frequently initiate pairing before both small chromosomes (ch7 and ch16) and large, gene-poor chromosomes (ch23 and chZ). Notably, no cells were observed with chZ pairing before any given autosomal chromosome, suggesting sex chromosome pairing may be delayed compared to autosome pairing.

Next, we sought to use our stripe paints to better understand the circular and forked chromosome configurations. In addition to identifying pachytene cells (Fig 3), we were able to identify cells in all stages of meiosis I up to metaphase I as well as cells in interphase and mitosis using the stripe paints (Figs 5A and S7). By examining chromosome configurations in zygotene nuclei, we found that chromosomes in the forked configuration pair at a single telomere domain (Fig 5B), while chromosomes in the circular configuration harbor pairing at both

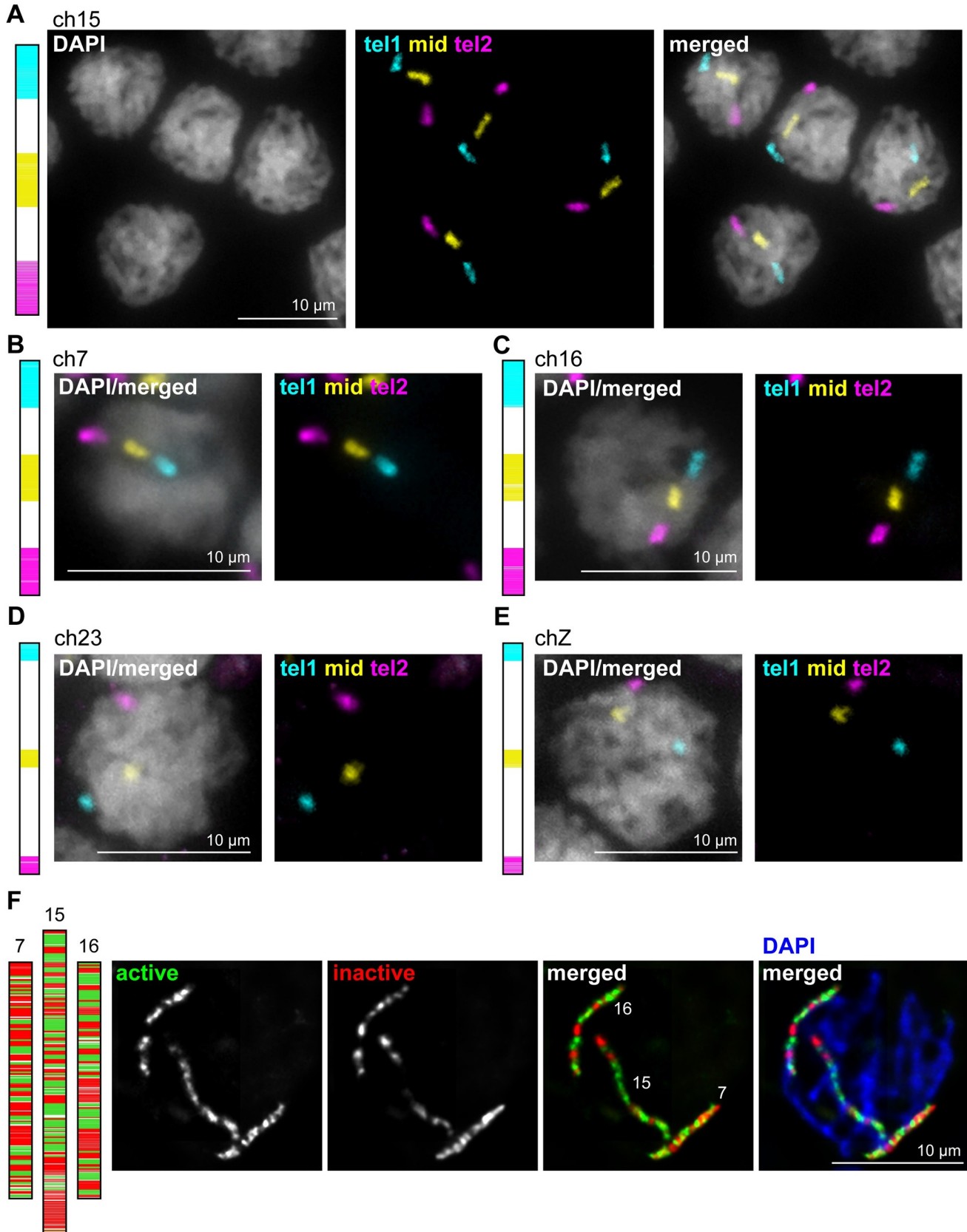

**Fig 3. Stripe and active/inactive chromosome paints in *B. mori* 5<sup>th</sup> instar testes squashes.** (A) Left: Schematic of stripe paints for ch15, with tel1 in cyan, mid in yellow, and tel2 in magenta. Right: Pachytene cells labeled with ch15 stripe paints. (B-E) Left: Schematic of stripe paints for ch7, ch16, ch23 or chZ. Right: Representative pachytene nucleus labeled with stripe paints shown on left. (F) Left: Schematic of active/inactive paints for ch7, 15, and 16. Active domains are shown in green, and inactive domains are shown in red. Right: Representative pachytene nucleus labeled with paints for all 3 chromosomes.

telomere domains but not the middle domain (Fig 5C and 5D). This telomere-telomere pairing in circular configurations includes both head-to-head pairing (tel1 paired with tel1 and tel2 paired with tel2, Fig 5C) as well as head-to-tail pairing (tel1 paired with tel2, Fig 5D), with head-to-head pairing being significantly more prevalent (Fig 5E).

The finding that both forked and circular chromosome pairing structures include paired telomere domains led us to predict that pairing may be initiated at chromosome ends. To test this hypothesis, we quantified the fraction of zygotene cells exhibiting pairing in only a single stripe domain (tel1, mid, or tel2) for four autosomes and chZ. In agreement with our prediction, we found a significant majority of cells harboring pairing at only one of the two telomere domains as opposed to pairing at only the middle domain (Fig 5F). Interestingly, some chromosomes showed a significant bias for pairing at one telomere domain versus the other, with ch15 pairing in the tel1 domain most often and ch23 pairing in the tel2 domain most frequently (Fig 5F). We investigated differences between the two chromosome ends on these chromosomes and found a significant bias in the distribution of genes along these chromosomes compared to the other chromosomes in our analyses, with gene-rich chromosome ends correlating with increased pairing initiation (Figs 5G and S8). This result, combined with our previous finding that large, gene rich chromosomes pair earlier, suggests that transcription may play an important role in pairing initiation in *Bombyx*. Interestingly, chZ does not follow this trend, as there is a slight bias in the distribution of genes toward the tel1 domain but a slight increase in pairing frequency of the tel2 domain. In agreement with our earlier finding that chZ pairs later than autosomes, this result suggests that sex chromosome pairing may be governed by different mechanisms than autosomal pairing, which may be transcription-based.

## Telomeres face poleward at random and recruit CENP-T at metaphase I in larval testes

When analyzing pairing initiation with our stripe paints, we also noted that traditional cruciform bivalents are formed at metaphase I. These bivalents are highly reminiscent of those seen in meiosis in the nematode *C. elegans*, with one telomere remaining paired and the other telomere facing poleward (Fig 5H, reviewed in [44]). While this "telokinetic" chromosome configuration was previously hypothesized to occur in *B. mori* meiosis [51–53], it has never before been directly observed. In *C. elegans*, either telomere on any given chromosome can harbor kinetochore activity, and both do so with equal probability, depending on where crossovers form during meiotic prophase I [42, 48]. Additionally, this telokinetic kinetochore activity is independent of the centromere-specifying histone CENP-A [46, 50, 54].

We first wanted to determine if both telomeres can also act as kinetochores at metaphase I in *B. mori*. One possibility is that the pairing bias we observed for prophase I persists through metaphase I such that gene-rich telomeres are more likely to remain paired and thus gene-poor telomeres are more likely to be functional kinetochores. To test this hypothesis, we analyzed metaphase I bivalents in larval testes squashes using our stripe paints for ch7, 15, 16, 23, and Z. Quantification of telomere orientation at the metaphase plate revealed that approximately half of metaphase I cells harbor pairing in the tel1 domain and half harboring pairing

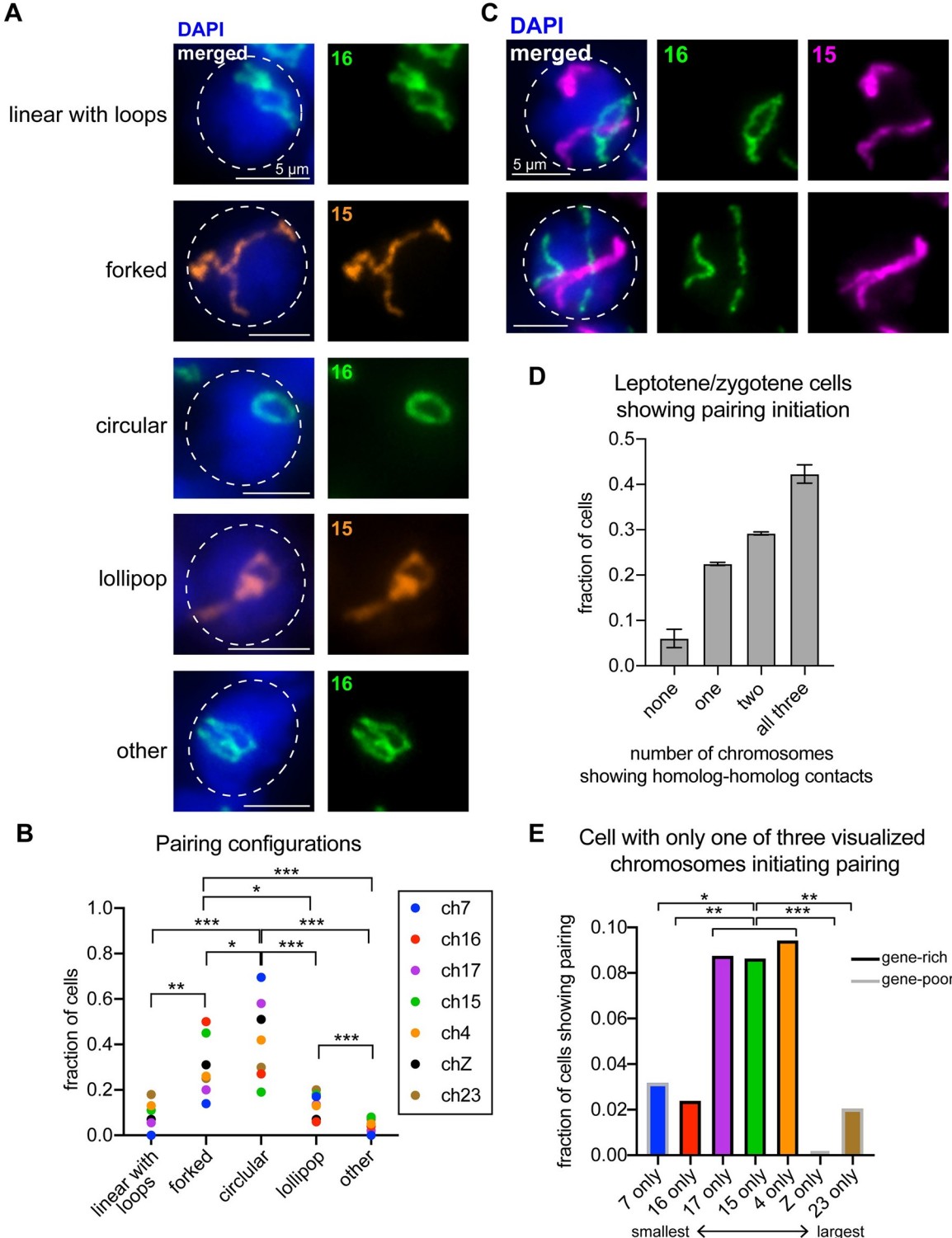

**Fig 4. Whole chromosome Oligopaints in zygotene-pachytene transition show a variety of folding and pairing configurations. (A)** Oligopaints for ch16 (green) or ch15 (orange) in representative zygotene nuclei showing the most prevalent partially paired chromosome configurations during pairing initiation. Dashed line approximates the nuclear edge. **(B)** Frequency histogram showing the fraction of cells harboring chromosomes in each partially paired configuration for the indicated chromosome. Statistics, Mann-Whitney test with each chromosome as a replicate. **(C)** Oligopaints for ch16 (green) and ch15 (magenta) in representative zygotene nuclei showing asynchronous pairing initiation. Top: Ch16 has begun pairing while ch15 remains entirely unpaired. Bottom: Ch15 is nearly

completely paired while ch16 remains unpaired. Dashed line approximates the nuclear edge. (**D**) Quantification of pairing initiation for three chromosomes at a time in cells with zygotene DAPI morphology and linear chromosome morphology. (**E**) Quantification of which chromosome is initiating pairing in nuclei where only one of the three visualized chromosomes is partially paired (while the other two chromosomes show no contact between homologs). Chromosomes are ordered smallest to largest. Gray outline indicates gene-poor chromosomes, while black outline indicates gene-rich chromosome. Statistics, Fisher's Exact Test comparing pairing initiated versus uninitiated. For all quantification, ***p<0.0001, **p<0.001, *p<0.01. All data were collected from n = 2–3 testes squashes per chromosome.

in the tel2 domain for all tested chromosomes, including chZ (Fig 5I). The unpaired telomere domain faces poleward, suggesting that like nematodes, *B. mori* telomere regions likely act as localized kinetochores during meiosis. This result indicates that, while pairing initiation shows a bias toward gene-rich telomeres, the orientation of chromosomes in the metaphase I bivalent is not biased, with both telomeres having an equal probability of either remaining paired or facing poleward.

Next, we repeated the experiment using ch7 stripe paints in whole mount late 5[th] instar larval testes to look at metaphase I telomere orientation in the native cellular context. This less invasive whole mount approach also allowed us to maintain critical spatial and temporal information and to better visualize germline development as a whole. As predicted based on our above and previous findings [53], in addition to meiotic cells, 5[th] instar larval testes also harbor mitotic cells that have unpaired homologs that highly resemble those seen in whole-mount embryos and testes squashes (Figs 6A–6C1, S7, and S9) and primary spermatocytes at all stages of meiosis I (Fig 6A–6D). Interestingly, *B. mori* and other Lepidopteran insects utilize two distinct spermatogenic pathways, ultimately resulting in apyrene sperm (without nuclei, which are thought to act as support cells for sperm migration [64]) and eupyrene sperm (with nuclei [64–70]). In whole-mount testes, we were able to clearly identify eupyrene secondary spermatocyte bundles (Fig 6D3) and mature eupyrene sperm (S10 and S11 Figs). Additionally, we identified secondary spermatocyte bundles apparently apyrene-destined, where some cells have no DNA, and the FISH signal is instead diffuse in the cytoplasm. This observation suggests that the cells in these bundles are beginning the process of nuclear degradation (Fig 6D4). Importantly, quantification of metaphase I bivalent formation in whole mount testes was completely in agreement with our findings from squashes, showing ch7 tel1 paired in 47% and ch7 tel2 paired in 53% of cells (Fig 6D2).

Finally, we wanted to interrogate how these bivalents attach to the meiotic spindle. While CENP-A is absent from the *B. mori* genome [55], mitotic kinetochore recruitment instead requires the DNA-binding protein CENP-T [34]. To determine if CENP-T is present on meiotic chromosomes or if microtubule attachment is independent of CENP-T, we performed co-immunofluorescence/FISH with anti-CENP-T and anti-Tubulin antibodies plus ch23 Oligopaint on dissociated cell cytospreads from late 4[th] instar larval testes. This cytocentrifuge-based approach allowed for clearer visualization of kinetochore/microtubule attachments, as chromosomes become more spaced during the centrifugation process (S12A and S12B Fig). FISH for a single chromosome enabled us to distinguish between meiotic and mitotic cells (one versus two FISH signals). This experiment revealed that indeed, CENP-T is present on meiotic chromosomes and therefore may facilitate attachment to microtubules. Interestingly, using super-resolution microscopy, we found that these kinetochores are more canonical than the cup-shaped kinetochores observed in *C. elegans* meiosis [50], with a single kinetochore structure forming at the microtubule-proximal end of each chromatid (S12C Fig). Together, these findings suggest that *B. mori* chromosomes form traditional bivalent structures at metaphase I with localized CENP-T-based kinetochore activity restricted to one telomeric region at random.

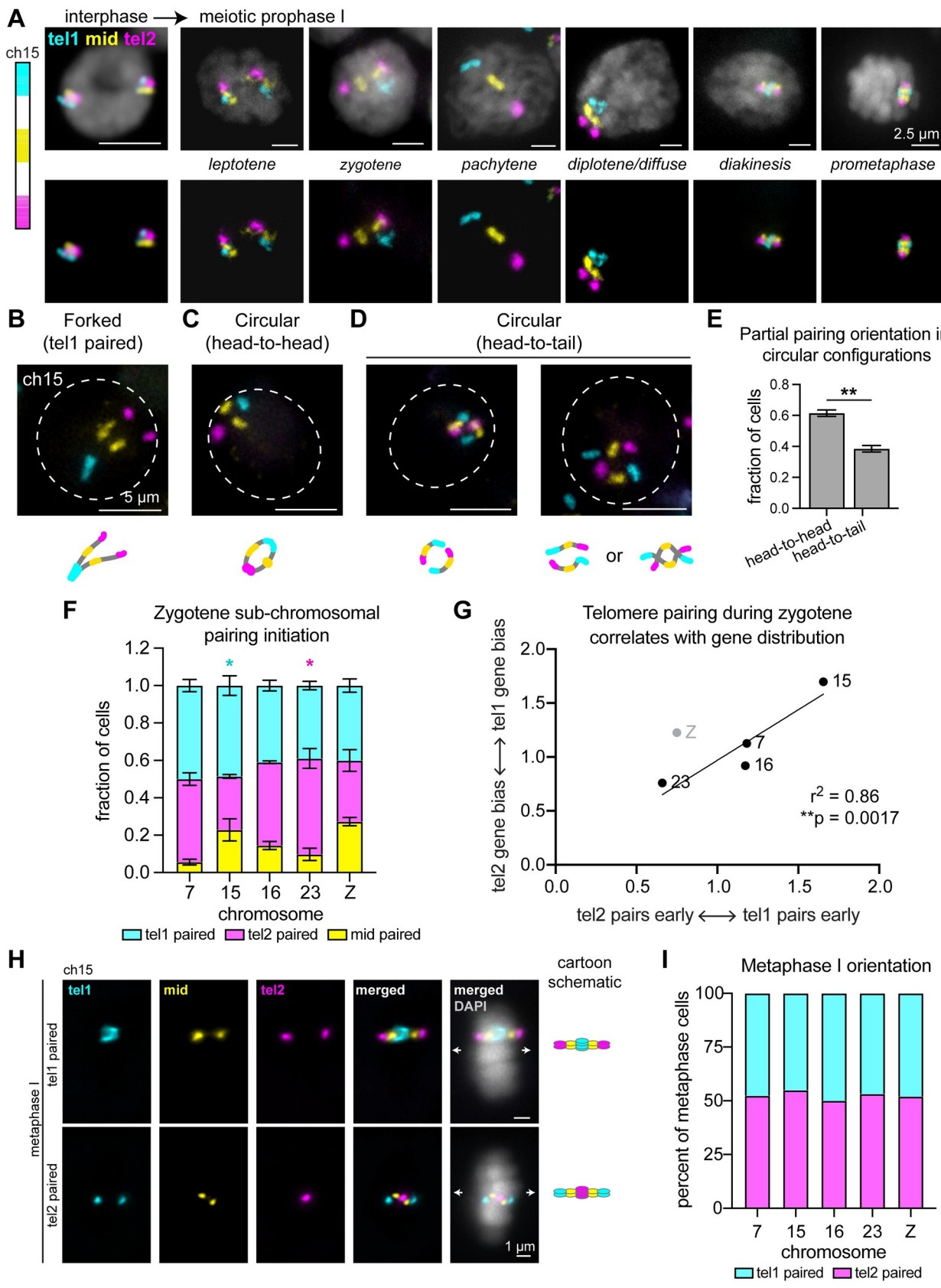

**Fig 5. Analysis of pairing and metaphase I bivalent formation in 5th instar larval testes squashes using stripe paints. (A)** Left: Schematic of stripe paints for ch15, with tel1 in cyan, mid in yellow, and tel2 in magenta. Right: representative nuclei at the designated stages labeled with ch15 stripe paints. When cells enter meiosis, chromosomes begin to decondense (leptotene) and homologs pair (zygotene). Pairing is complete by pachytene, with complete synapsis for crossing over, and chromosomes are linear. Chromosomes begin to condense for segregation in diplotene and diakinesis. DAPI is shown in gray. **(B)** Top: Representative zygotene nucleus labeled with ch15 stripe paints showing a forked chromosome pairing configuration with tel1 paired. Bottom: Cartoon schematic of ch15 in above cell. **(C)** Top: Representative zygotene nucleus labeled with ch15 stripe paints showing a circular chromosome with a head-to-head pairing configuration. Bottom: Cartoon schematic of ch15 in above cell. **(D)** Top: Two representative zygotene nuclei labeled with ch15 stripe paints showing a circular chromosome with a head-to-tail pairing configuration. Bottom: Cartoon schematics of ch15 in above cells. **(B-D)** Dashed line indicates nuclear edge. **(E)** Quantification of circular chromosome pairing configurations in zygotene. Graph includes pooled data for ch15 and ch23, with n = 3 testes quantified for each. Error bars show standard deviation between chromosomes. $^{**}p = 0.0082$, unpaired t-test. **(F)** Quantification of stripe domain pairing in cells with only one stripe domain paired. n = 3 testes were quantified for each chromosome. For all chromosomes, pairing at both telomere domains is significantly enriched compared to the middle domain ($p < 0.0001$, Fisher's exact test comparing tel1 vs mid and tel2 vs mid). Cyan asterisk = tel1 significantly enriched compared to tel2, magenta asterisk = tel2 significantly enriched compared to tel1 (Fisher's exact test, $p < 0.05$). **(G)** Dot plot showing tel1:tel2 pairing ratio (X-axis) versus tel1:tel2 gene density ratio (Y-axis). Line of best fit calculated for autosome data only. Z chromosome data (gray) included for comparison. **(H)** Metaphase I bivalents labeled with ch15 stripe paints. Top: bivalent with pairing in tel1 domain. Bottom: bivalent with pairing in tel2 domain. Schematics of bivalents shown on the right. **(I)** Quantification of metaphase I orientation for ch7, 15, 16, 23, and Z. Ch7, n = 357 (48% tel 1 paired). Ch15, n = 182 (45% tel 1 paired). Ch16, n = 252 (50% tel1 paired). Ch23, n = 519 (47% tel1 paired). ChZ, n = 169 (48% tel1 paired). Each FISH assay was performed on n = 2–3 testes.

## Homolog pairing at chromosome ends is persistent in *B. mori* female larval ovaries

In contrast to what we and others have observed in *B. mori* males, homolog pairing in females is reported to be unconventional, without chiasma formation and with the SC partially breaking down and transforming into a large proteinaceous mass greater than one micron in width between the homologs before metaphase I [53, 56–58, 71]. Despite these previous observations, we found a significant number of nuclei with homologs paired end-to-end in late 4th/early 5th instar larval ovary squashes (Figs 2 and S4). This finding led us to wonder whether homolog pairing is more stable in *B. mori* female meiosis than previously appreciated.

To better visualize homolog pairing and the linear progression of female meiosis, we performed whole-mount DNA FISH with ch7 stripe paints in 5th instar larval ovaries and used super-resolution microscopy to measure the distance between homologous copies. Like *Drosophila*, the *B. mori* larval ovary is composed of polytrophic meroistic ovarioles (where the oocyte is part of a group of inter-connected cells including supportive nurse cells). Each ovariole contains a linear arrays of developing egg chambers, with the tip (germarium) harboring germline stem cells that are mitotically dividing and the most mature chambers being the most distal from the stem cell niche ([63, 72, 73], Figs 7A and S13). However, as moths have a much shorter adult lifespan than flies (only 5–7 days for silkmoths versus 2–3 months for *Drosophila*), the majority of oogenesis occurs in the larval and pupal rather than adult stage [63]. Therefore, 5th instar larval ovaries harbor oocytes in all stages of pairing.

Interestingly, despite our diffraction-limited imaging of ovary squashes appearing to show tight homolog pairing in early pachytene and late pachytene (Figs 2, 7, and S4), super-resolution imaging of homologs in the zone of pairing in the germarium (the early stages of pachytene) revealed that homologous chromosome copies are approximately 0.15–0.20 μm apart at this stage (Fig 7B and 7E). This configuration is a much looser pairing association than we observed in larval testes, which harbor closely paired homologs with no detectable separation even at super-resolution (S14 Fig). By late pachytene, when nurse cells and the developing oocytes begin to differentiate, ch7 homologs are even further separated (0.25–0.30 μm, Fig 7C and 7E). In the first egg chambers most proximal to the germarium, chromosomes are still pachytene-like in structure, but homologs are approximately 0.30–0.35 μm apart (Fig 7E). The majority of oocytes in the developing egg chambers outside the germarium in 5th instar larvae are arrested in late diakinesis [56], which should be after transformation of the SC at the end of

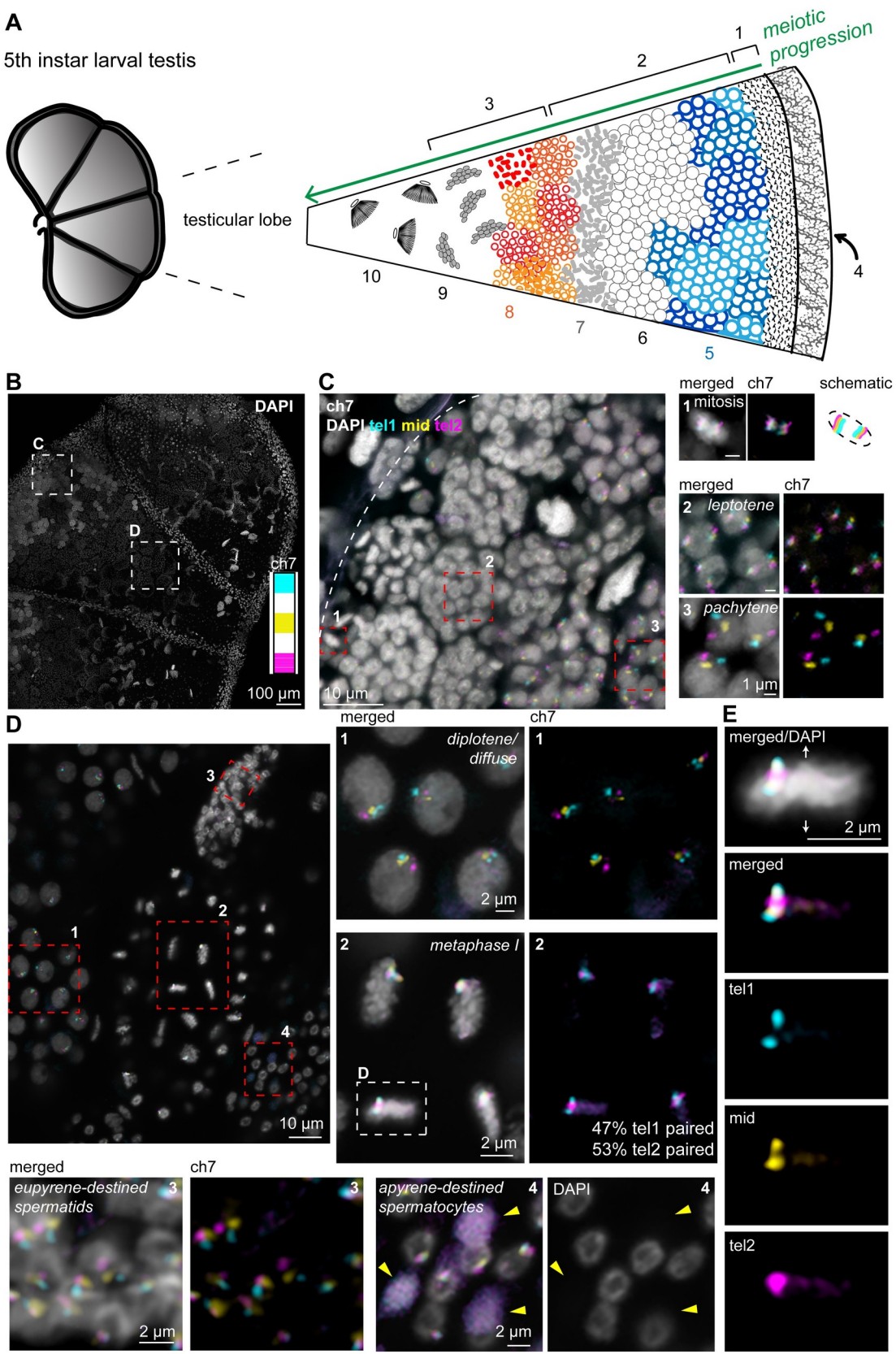

**Fig 6. FISH with stripe chromosome paints in whole mount 5<sup>th</sup> instar larval testes. (A)** Cartoon schematic of 5[th] instar larval testis. Mature 5[th] instar larval testes are comprised of four testicular lobes, each of which harbors germline stem cells (1, mitotic zone) and spermatocytes in all stages of meiosis up to mature sperm; progressing from right to left in the image as indicated: (2) meiosis I/primary spermatocytes, (3) meiosis II/secondary spermatocytes. Additionally, each lobe is surrounded by somatic cells in the sheath (4) and in the septae separating the lobes. (5–6) prophase I: (5) leptotene/zygotene, (6) pachytene/diplotene, (7) metaphase/anaphase I, (8) prophase/metaphase II, (9) spermatids, (10) mature sperm bundles. **(B)** Confocal image of 5[th] instar larval testis stained with DAPI. Boxes indicate subsequently zoomed panels as indicated. Inset: ch7 stripe paints used in C-E. **(C)** Zoom of mitotic and pachytene region of larval testes as shown in A, labeled with ch7 stripe paints. Red boxes indicate zooms shown to the right. **(C1)** mitotic cells–chromosomes condensed, homologs unpaired, and aligned at the metaphase plate. Note how chromosomes are compacted perpendicular to the metaphase plate. **(C2)** leptotene cells, chromosomes are slightly decondensed and homologs are unpaired. **(C3)** pachytene cells, chromosomes are paired head-to-tail and linear. **(D)** Zoom of late prophase/metaphase I region of larval testis as shown in A, labeled with ch7 stripe paints. Red boxes indicate zooms shown to the right and below. Please note that some FISH signal background is visible within the DNA in this image. This is an artifact of the size of the larval gonad and the imaging depth. **(D1)** diplotene cells (diffuse stage), chromosomes are still paired and beginning to condense. Left: merged with DAPI in gray. Right: ch7 tel1 in cyan, mid in yellow, and tel2 in magenta. **(D2)** metaphase I cells labeled with ch7 stripe paints. White box indicates zoom shown in E. Left: merged with DAPI in gray. Right: tel1 (cyan) and tel2 (magenta) paints. Percent of cells with tel1 or tel2 paired indicated inside the panel. Cells from 1–2 bundles each from n = 3 testes were quantified for a total of n = 196 cells (47% tel1 paired). **(D3)** eupyrene-destined secondary spermatocyte bundle. Left: merged ch7 stripe paints with DAPI in gray. Right: ch7 stripe paints, tel1 (cyan) and tel2 (magenta). **(D4)** apyrene-destined secondary spermatocyte bundle. Left: merged ch7 stripe paints with DAPI in gray. Right: DAPI. Yellow arrowheads indicate spermatocytes that have already undergone nuclear degradation. Scale bar = 2 μm for all panels 1–4. **(E)** Zoom of metaphase I cell indicated in C2. Top: merged with DAPI in gray. Bottom: ch7 tel1 (cyan), mid (yellow), and tel2 (magenta) paints. Bivalent pairing is in tel2 domain in this cell. Note: zoomed fields for all panels may display a slightly different Z position than the larger field views for better clarity.

pachytene. In agreement with this hypothesis, super-resolution imaging revealed that in oocytes in the most mature egg chambers most distal from the germarium, homologs remain linear but are approximately 0.45–0.55 μm apart, with the two homologous copies connected only at telomere domains, forming a large chromosomal loop (Fig 7D–7E and S1 Movie). This finding suggests that in the absence of chiasma and after the partial breakdown and transformation of the SC, contacts between homologous chromosomes persist throughout meiotic prophase I in female *B. mori*. These contacts appear to be mediated by both telomeric pairing mechanisms possibly in addition to any remaining modified SC, revealing that homologs are indeed closer together than the predicted one micron separation.

## Discussion

The silkworm, *B. mori*, was a model system for studying meiotic chromosomes for decades. Like *Drosophila*, silkworms are readily reared in a laboratory setting and amenable to genetic manipulation including RNAi and CRISPR. However, unlike fruit flies, silkworms are large in size, combining the increased ease of dissection and structure visualization commonly associated with mammalian models with the short generation time of an insect system. The large amount of tissue provided by *B. mori* increases the feasibility of genomics assays and other cell population-based approaches, which generally require a large number of cells. Importantly, the recent sequencing of the *B. mori* genome revealed that there is a high degree of sequence homology between silkworm genes and mammalian disease genes [60, 74–76]. Furthermore, *B. mori* harbor 28 chromosomes while humans have 23 (*Drosophila* have only 4), and our studies along with others have illustrated that, like in mammals, somatic homolog pairing is largely absent in *B. mori* [77]. This observation is in stark contrast to the high levels of somatic homolog pairing seen in *Drosophila* [78, 79], making the study of *B. mori* genome dynamics more directly relevant to human biology.

Our studies examine pairing initiation in great detail in larval testes. We found that pairing is asynchronous in *B. mori*, with long, gene-rich chromosomes more frequently initiating pairing before their shorter or gene-poor counterparts. Interestingly, in zygotene nuclei, we observed a wide variety of partially paired chromosome configurations, many of which include

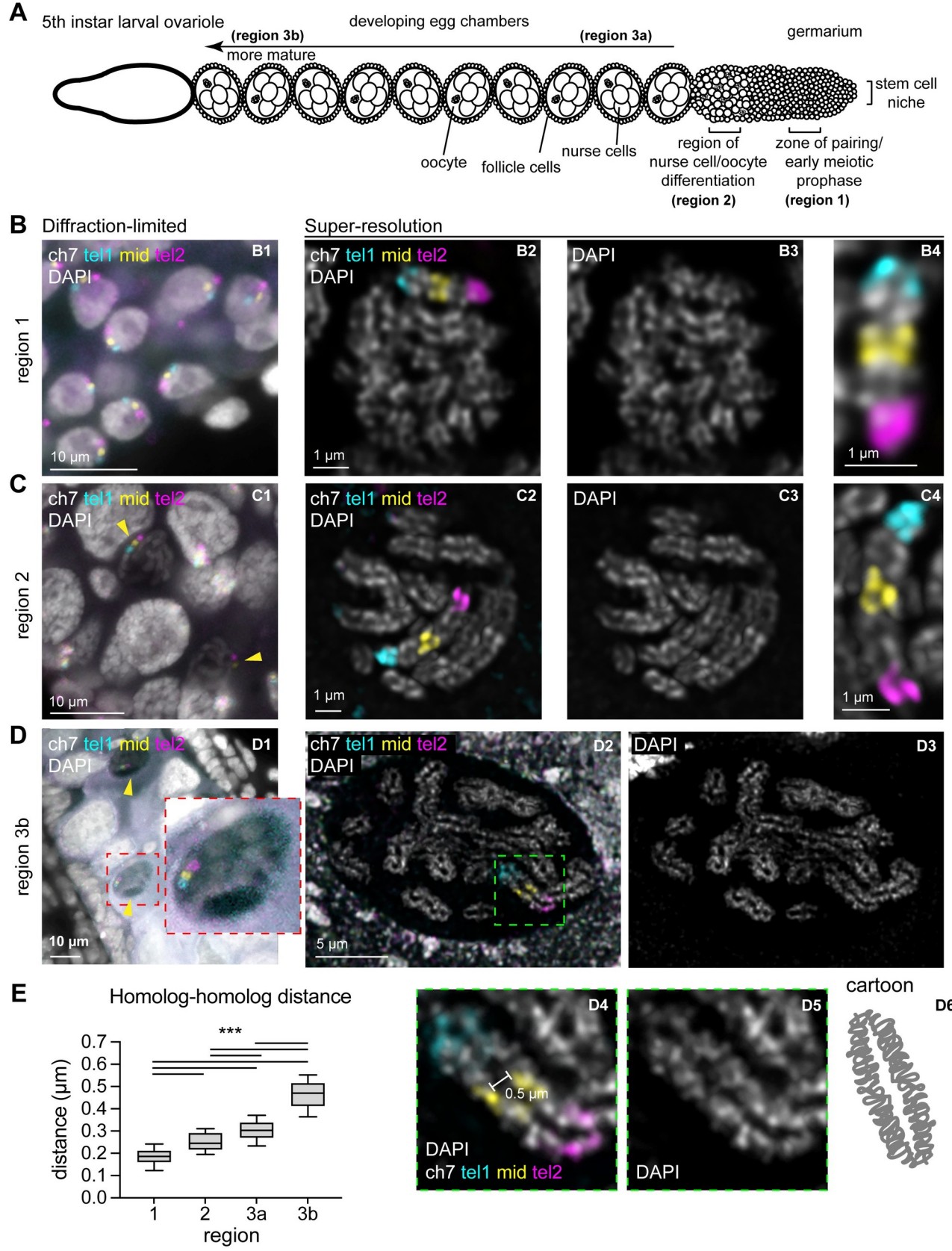

**Fig 7. FISH with stripe chromosome paints in whole mount 5ᵗʰ instar larval ovary. (A)** Schematic of ovariole from 5ᵗʰ instar *B. mori* larval ovary. **(B1-D1)** Representative field of nuclei from larval ovary labeled with ch7 stripe paints imaged with diffraction-limited confocal microscopy. DAPI is shown in gray. **(B)** germarium (zone of pairing, region 1), **(C)** region of differentiation (region 2), **(D)** mature larval oocytes (region 3b). **(C1-D1)** yellow arrowheads indicate oocytes which were identified based on their unique chromosome morphology and weak DAPI stain. **(B2-D5)** Representative nuclei labeled with ch7 stripe paints and DAPI imaged with super-resolution microscopy. **(B4-D4)** zoom of ch7 shown in B2-D2. **(D5)** zoom of ch7 shown in D2, DAPI only. **(D6)** cartoon schematic of chromosomes in mature larval oocytes. **(E)** Measurement of distances between homologs obtained from super-resolution images. n = 3 ovaries were imaged and quantified. Measurements were taken from 3–5 chromosomes in 5–10 cells per ovary, resulting in 50–75 data points per region. ***p<0.001, Mann-Whitney test.

loop-like structures. We suspect that some of these loops (such as the "linear with loops" configuration) represent interlocked chromosomes as previously observed in *Bombyx* [80, 81], while others, like the highly abundant "circular" and "forked" configurations, represent pairing intermediates. It is also worth noting that "pairing" as measured by Oligopaint signal colocalization, even at approximately 140 nm x-y super-resolution, may not necessarily indicate SC formation and true synapsis (and we currently do not have SC antibodies for *B. mori*). We believe it is possible, and even likely, that there is a chromosome-wide "search" leading to transient colocalization of various chromosome regions (including the middle domain of chromosomes), but SC formation may always be initiated at telomeres, as in *C. elegans* [48, 82–84].

Importantly, we found that chromosome ends more frequently pair before the central domain of both autosomes and chZ. Additionally, we observed a bias toward gene-rich chromosome ends initiating pairing earlier than gene-poor chromosome ends of autosomes. Together, these data support a model in which transcription, the transcription machinery, or active histone modifications modulate pairing initiation in *Bombyx*. This study is not the first report linking transcription with homolog pairing. In fact, somatic pairing in *Drosophila* is largely influenced by transcription, and furthermore, transcription factories have been hypothesized to facilitate meiotic pairing in other species [85–87]. Still, whether or not telomeric genes are even transcribed in meiosis is unclear as meiotic transcriptome profiles for *B. mori* have yet to be established, and thus, the exact mechanism behind pairing initiation in *Bombyx* and beyond remains unclear.

Our FISH-based approach further reveals how chromosomes compact after prophase I and partially unpair when aligned at metaphase I. While crossing over has been reported in male meiosis in *B. mori* [52, 56], clear chiasmata were not apparent in post-pachytene spermatocytes (Figs 2, 4, and 5). We suspect this inability to detect chiasmata is likely due to the small size and compact nature of *B. mori* chromosomes in diplotene. We also show that mitotic chromosomes in *B. mori*, which are holocentric in structure, align parallel to the metaphase plate, with both telomeres aligned with the plate and homologs remaining unpaired. Our studies further illustrate that, like those in *C. elegans*, *B. mori* chromosomes do not retain the holocentric configuration in meiosis. Instead, meiotic chromosomes at metaphase I in spermatogenesis align perpendicular to the metaphase plate such that telomeric regions face the spindle poles, recruit CENP-T, and act as localized kinetochores. Super-resolution imaging of meiotic chromosomes at metaphase I revealed that these CENP-T-based kinetochores do not resemble the cup-shaped kinetochores in *C. elegans* [50], but instead appear to be more canonical in structure, with a single kinetochore forming at the telomere of each chromatid.

Moreover, we demonstrate that both telomeres are equally likely to face poleward and harbor kinetochore activity. A similar telokinetic mechanism for meiosis has also been observed in the holocentric milkweed bug *Oncopeltus fasciatus* [88] and the kissing bug *Triatoma infestans* [89]. Whether crossover position dictates bivalent structure in *B. mori* or other holocentric insects, as in *C. elegans* [42, 45, 48–50], remains to be explored. Interestingly, the broadly conserved centromere-specific histone H3 variant CENP-A is absent from the genome

of Lepidopteran insects [55], suggesting that both mitotic and meiotic chromosome segregation in *Bombyx* occur independently of CENP-A.

We combined our Oligopaints with super-resolution microscopy to measure chromosome-wide pairing in *B. mori* female meiotic prophase. We show that homologs are never paired as tightly in female meiosis as they are in male meiosis, in agreement with lack of chiasma formation in ovaries [53, 56, 57]. Additionally, we find that after transformation of the SC at the end of pachytene, homologs appear to remain linked through telomere regions. Previous studies also suggest that modified SC remaining at this stage may play a role in keeping homologs together until anaphase I [58]. Yet, the exact mechanism facilitating and maintaining pairing in female meiosis in *B. mori* remains to be elucidated. Finally, the studies presented here illustrate that Oligopaints can be designed using, and act as validation for, draft genome assemblies, and demonstrate the feasibility of using Oligopaints to study meiotic chromosomes. Using this approach, we reveal novel insights into pairing initiation and meiotic chromosome segregation that may be conserved across species.

## Materials and methods

### *B. mori* strains and cell line

Embryos were obtained from Carolina Biological (Burlington, NC), Coastal Silkworms (Jacksonville, FL), Mulberry Farms (Fallbrook, CA), or were freshly laid in the lab by adults derived from embryos from these sources. Some larvae were obtained from Rainbow Mealworms (Compton, CA). Embryos were kept at 4˚C for less than 1 mo. For rearing, embryos were transferred to 28˚C, and larvae were fed fresh mulberry leaves or powdered mulberry chow (Carolina Biological or Rainbow Mealworms). BmN4 cells are commercially available from ATCC (Manassas, VA).

### Oligopaint design and synthesis

Oligopaint libraries were designed as described in the main text. Active and inactive domains were determined primarily based on CENP-T depletion or enrichment. CENP-T ChIP-seq profiles were obtained from BmN4 cells, and domains were called as previously described [37] with the following modifications: CENP-T ChIP-seq signal originally in 10 kb windows was averaged over 50 kb. Subsequently, negative CENP-T domains were subtracted from positive CENP-T domains to obtain final CENP-T depleted domains. As previously observed, domains enriched for CENP-T were shown to strongly correlate with enrichment for the repressive histone mark H3K27me3, while domains depleted of CENP-T were shown to strongly correlate with enrichment of the active chromatin marks H3K4me3 and H3K36me3. All information regarding genomic coordinates for Oligopaints and probe density can be found in Tables 1–5. Oligo pools were purchased from CustomArray/GenScript (Redmond, WA; ch 7, 15, 16) or Twist Biosciences (San Francisco, CA; ch 4, 17, 23, Z). Oligopaints were synthesized as previously described by adding barcodes to each oligo for PCR-based amplification [17, 30, 90].

### Preparation of meiotic squashes and DNA FISH

For meiotic squashes, late 4th instar or early 5th instar larvae (approximately 3 in long) were sacrificed by decapitation. The caterpillars where then cut open anterior to posterior and fileted on a silicone dissecting dish using standard sewing needles. Gonads were harvested using forceps and placed into 1.5 mL tubes containing sf-900 (Gibco/ThermoFisher, Waltham, MA) tissue culture media. Gonads were then rinsed thrice in 1X PBS, then incubated in 0.5% sodium citrate for 8–10 min. Using forceps, gonads were then transferred to siliconized

coverslips (1 gonad per coverslip) and covered with ~10 μL of 45% acetic acid/1% PFA/1X PBS and fixed for 6 min. Using a poly-L-lysine coated glass slide, gonads were then physically squashed and slide/coverslip were flash frozen in liquid nitrogen. After carefully removing slides from liquid nitrogen, coverslips were removed with a razor blade, and slides were post-fixed in cold (pre-chilled to -20˚C) 3:1 methanol:glacial acetic acid for 10 min. After fixation, slides were washed thrice in 1X PBS and subjected to an ethanol row at -20˚C (70%, 90%, 100% ethanol, 5 min each) before drying completely at room temp. Slides were dried for 24–72 h.

FISH on meiotic squashes was performed as previously described for mitotic spreads [31]. Briefly, after drying slides, slides were denatured at 72˚C for 2.5 min in 2xSSCT/70% formamide before again drying with an ethanol row at -20˚C. Slides were then left to air dry for 10 min at room temperature. Primary Oligopaint probes were resuspended in hybridization buffer (10% dextran sulfate/2xSSCT/50% formamide/4% polyvinylsulfonic acid), placed on slides, covered with a coverslip, and sealed with rubber cement. Slides were denatured on a heat block in a water bath set to 92˚C for 2.5 min, after which slides were transferred to a humidified chamber and incubated at 37˚C overnight. The next day, coverslips were removed using a razor blade, and slides were washed as follows: 2×SSCT at 60˚C for 15 min, 2×SSCT at RT for 15 min, and 0.2×SSC at RT for 5 min. Fluorescently labeled secondary probes were then added to slides, again resuspended in hybridization buffer, covered with a coverslip, and sealed with rubber cement. Slides were incubated at 37˚C for 2 h in a humidified chamber before repeating the above washes. All slides were stained with DAPI and mounted in Prolong Diamond (Invitrogen/ThermoFisher, Waltham, MA). Slides were cured overnight before sealing with clear nail polish and imaging.

## FISH on whole-mount gonads and embryos

For whole-mount DNA FISH in gonads, ovaries and testes from late 5th instar larvae (after cessation of eating) were dissected in sf-900 cell culture media, washed thrice briefly with 1X PBS, and then fixed for 30 min in 4% PFA in PBS with 0.1% Triton-X-100 (0.1% PBS-T) at RT. Gonads were then washed again thrice in 1X PBS and permeabilized with 0.5% PBS-T for 15 min at RT. Gonads were pre-denatured by washing as follows: 2xSSCT for 10 min at RT, 2xSSCT/20% formamide for 10 min at RT, 2xSSCT/50% formamide for 10 min at RT, 2xSSCT/50% formamide for 3 h at 37˚C, 2xSSCT/50% formamide for 3 min at 92˚C, 2xSSCT/ 50% formamide for 20 min at 60˚C. To the 2xSSCT/50% formamide, 100 pmol of each probe was directly added. Gonads were then denatured for 3 min at 92˚C and incubated overnight at 37˚C. The next day, gonads were washed: 3x 30 min each in 2xSSCT/50% formamide at 37˚C, 1x 15 min in 2xSSCT at RT. 20 pmol of each secondary oligo was added with 50% formamide and incubated for 3 h at 37˚C. Final washes were performed (2x 30 min in 2xSSCT/50% formamide at 37˚C, 1x 10 min in 2xSSCT/50% formamide at RT, 1x 10 min in 2xSSCT/20% formamide at RT, 1x 10 min in 2xSSCT at RT), gonads were stained with DAPI, and mounted on slides with Prolong Diamond (Invitrogen/ThermoFisher, Waltham, MA).

For whole-mount embryo FISH, diapausing embryos were removed from 4˚C and kept at RT for 3–5 d. Chorions were weakened by soaking in 50% bleach for 15 min and then manually removed with forceps. Embryos were subsequently fixed for 30 min in 4% PFA in 0.1% PBS-T at RT, and FISH was performed as described above for whole-mount gonads.

## IF-FISH on cytocentrifuge testes spreads

Testes were dissected as described above in sf-900 tissue culture media, followed by three quick washes in 1X PBS, then incubated in 500 μL 0.5% sodium citrate for 8–10 min in 1.5 mL

tubes. Gonads were then gently crushed with a p1000 pipet tip, and cells were dissociated by gently pipetting up and down. 250 µl of the dissociated cells were added to a cytofunnel containing a poly-L lysine-coated slide. Cytocentrifugation (Shandon Cytospin 4, ThermoFisher, Waltham, MA) was performed for 5 min at 500 rpm with high acceleration. Slides were then fixed in 4% PFA for 10 min followed by three washes in 1X PBS. Cells were subsequently permeabilized in 100% methanol at -20˚C for 20 min, and then in 0.5% PBS-T for 15 min at RT. IF was then performed by blocking slides in 5% milk for 1 h at RT followed by incubation with primary antibodies diluted 1/500 in 5% milk overnight at 4˚C. Custom anti-CENP-T [34] and commercial anti-Tubulin antibodies were used (MilliporeSigma T6074, Burlington, MA).

The next day, slides were washed thrice in 0.1% PBS-T followed by incubation with secondary antibodies diluted 1/500 in 5% milk (goat anti-mouse alexa 546 and goat anti-rabbit alexa 488, ThermoFisher). After antibody incubation, slides were washed 3X in 0.1% PBS-T and post-fixed with 4% PFA for 10 min before proceeding with FISH as previously described [30].

## Meiotic staging

Stages of meiosis were determined based largely on DAPI morphology, chromosome morphology, and/or cell position in whole-mount gonads. Leptotene/zygotene chromatin (DAPI) is a dense lampbrush structure, with chromosomes being linear but unpaired in leptotene and linear but partially paired in zygotene. Pachytene chromatin is a loose lampbrush with chromosomes being linear and completely paired from end to end. Diplotene/diakinesis chromatin is diffuse with chromosomes being paired or partially paired, nonlinear, and condensing. Metaphase I chromatin is highly condensed and aligned at the metaphase plate, with chromosomes being paired in a bivalent structure and highly condensed. Late 4th-early 5th instar male larvae were used for squashes as they only possess primary spermatocytes in meiosis I. In whole-mount testes, meiosis I and II were distinguished based on position in the gonad and based on the number of cells per bundle (with meiosis I bundles harboring approximately 64 cells and meiosis II bundles harboring approximately 128 cells).

## Mitotic spreads from BmN4 cells

To induce mitotic arrest, approximately $1 \times 10^5$ cells were treated with 0.5 µg/mL Colcemid Solution (Gibco/ThermoFisher) for 2 h in a 28˚C heat block. Cells were then spun for 5 min at 600x $g$ at RT to pellet and resuspended in hypotonic solution (500 mL of 0.5% sodium citrate). Cells were incubated in hypotonic solution for 8 min at RT. 100 µL of the cell suspension was then placed in a cytofunnel and spun at 1200 rpm for 5 min with high acceleration using a cytocentrifuge (Shandon Cytospin 4; ThermoFisher). For FISH, slides were then fixed in cold 3:1 methanol: acetic acid for 10 min and washed 3X 5 min in PBS-T (PBS with 0.1% Triton X-100). FISH was performed as described above for meiotic squashes.

## Imaging, quantification, and data analysis

Widefield images of meiotic squashes and cytocentrifuge meiotic cells were acquired on a Leica DMi6000 wide-field inverted fluorescence microscope using an HCX PL APO 63x/1.40–0.60 Oil objective (Leica Biosystems, Buffalo Grove, IL), Leica DFC9000 sCMOS Monochrome Camera, and LasX software. Whole mount images were acquired using a Zeiss LSM 780 point scanning confocal (Zeiss Microscope Systems, Jena, Germany) with high sensitivity 32 anode Hybrid-GaAsP detectors. BmN4 mitotic spreads were acquired on a Zeiss AxioObserver Z1 wide-field inverted fluorescence microscope with 100x/1.4 oil Plan-APO objective, a Hamamatsu C13440 ORCA-Flash 4.0 V3 Digital CMOS camera, and ZEN blue software.

Super-resolution imaging of cytospin meiotic spindles and CENP-T immunofluorescence was performed using a Nikon CSU-W1 SoRa spinning disk confocal microscope (Nikon Instruments, Inc, Melville, NY), controlled by Nikon Elements AR software ver 5.21.02. Images were collected using z-stacks of multi-channel fluorescence. The image detectors used were dual Hamamatsu Fusion BT (model C14440-20UP) cameras. SoRa super-resolution mode was used with objective lens Nikon CFI Apo TIRF 60x/1.49 Oil DIC N2, and intermediate SoRa magnification 4.0x, producing a pixel size of 27 nm. The main excitation/emission dichroic beamsplitter used was a Semrock Di01-T405/488/568/647 (IDEX Health & Science, LLC, Rochester, NY), and the image-splitter dichroic was 'DM A561 LP'. All emission filters were made by Chroma Technology Corporation (Bellows Falls, VT). Alexa Fluor 647 was excited by the 640 nm laser and emission filtered by ET655lp. Alexa Fluor 546 was excited by the 561 nm laser and emission filtered by ET605/52m. Alexa Fluor 488 was excited by the 488 nm laser and emission filtered by ET520/40m. DAPI was excited by the 405 nm laser and emission filtered by Semrock FF02-447/60. Post-processing of the images was performed in Nikon Elements AR software and included denoising via Denoise.ai and 3D deconvolution in automatic mode. Super-resolution imaging of Oligopaints in whole mount ovaries was performed using a Zeiss LSM 880 Airyscan with 63x/1.4 oil Plan-APO objective. Airyscan images were acquired in SR mode using the Zeiss 32 channel Airyscan detector, and images were processed under auto strength setting with Zen black software. Leica and Zeiss-acquired images were processed using Huygens deconvolution software (SVI, Hilversum, Netherlands). For all images, tiffs were created in ImageJ. All quantification was performed manually. All data are reported in S1 File.

## Supporting information

**S1 Fig. ChIP-seq profiles used to design active and inactive chromosome paints.** Screenshots of ChIP-seq data used to design active/inactive chromosome paints. Inactive: H3K27me3 (orange), Centromere Protein T (CENP-T; dark red). Called inactive paint domains shown in bright red. Active: H3K36me3 (teal), H3K4me3 (dark green). Called active paint domains shown in bright green. ChIP-seq data were previously published (see Materials and Methods). Chromosome 7 is shown in A, part of chromosome 15 is shown in B (coordinates Chr15:3,900,000–12,891,000), and chromosome 16 is shown in C.
(TIF)

**S2 Fig. Mitotic and meiotic nuclei in early 4th-5th instar larval testis squash.** Representative image showing a cluster of mitotic chromosomes (yellow arrow, left) and a cluster of meiotic prophase I cells (green arrow, right) labeled with DAPI.
(TIF)

**S3 Fig. Somatic cells with unpaired homologs from 4th-5th instar larval testis squash.** Three nuclei (DAPI in gray) labeled with Oligopaints for ch7 (magenta), ch15 (orange), and ch16 (green) in somatic cells from late 4th instar larval testes gonad squashes. Two signals per nucleus indicate unpaired homologs.
(TIF)

**S4 Fig. Mitotic cell labeled with whole chromosome paints from 4th-5th instar larval testis squash.** Representative mitotic cell labeled with whole chromosome paints for ch4 (blue), ch7 (magenta) and ch17 (red).
(TIF)

**S5 Fig. Post-pachytene early differentiating oocytes and nurse cells from 4th-5th instar larval ovary squash labeled with whole chromosome paints.** Top: Representative field from

early 5th instar larval ovary squash labeled with whole chromosome paints for ch17 (red), ch23 (cyan), and chZ (yellow). Left, merged with DAPI, right, paints alone. Yellow arrowheads indicate early differentiating oocyte cells. Note: female *B. mori* are heterogametic and harbor a single Z and a single W chromosome (compared to two copies of ch17 and ch23). Bottom: zoom showing early differentiating oocyte and nurse cells, as indicated. As both oocytes and nurse cells undergo homolog pairing in early meiotic prophase, homologs remain in close proximity in all early differentiating nurse cells.
(TIF)

**S6 Fig. Mitotic spreads from BmN4 cultured cells labeled with whole chromosome Oligopaints.** Left: representative mitotic chromosome spread from BmN4 cultured cells labeled with whole chromosome paints for ch7 (orange), ch15 (red), and ch16 (green). Gray dashed box indicates zoom shown to the right. No entire chromosomes are labeled in the cell line, and instead, several chromosomes are partially labeled with each paint, indicating a large number of translocations in this cell line compared to animals.
(TIF)

**S7 Fig. Mitotic cell from 4th-5th instar larval testis squash.** Left: schematic of ch15 with stripe paints. Middle two panels: Representative nucleus at metaphase of mitosis labeled with ch15 stripe paints. Arrows indicate the direction of the spindle poles. Right: Cartoon representation of nucleus in middle panels.
(TIF)

**S8 Fig. Gene density distribution across relevant chromosomes.** Screenshots of genome browser views of ch7, 15, 16, 23, and Z. Top: ChIP-seq profiles for active histone modifications from BmN4 cells: H3K36me3 (green), H3K4me3 (blue). Middle: gene positions. Bottom: tel1, mid, and tel2 Oligopaint domains. Note: ch15 tel2 is depleted of genes and active histone marks compared to tel1, while ch23 tel2 is enriched for active marks and genes compared to tel1.
(TIF)

**S9 Fig. Ch15 stripe paints in a post-diapause whole mount embryo.** (A) Whole-mount embryo stained with DAPI imaged at 10x. (B) 100x image of whole-mount embryo stained with DAPI (top) and labeled with ch15 stripe paints (bottom) showing representative mitotic and interphase somatic cells. Note: both 10x and 100x images show Z-projections of multiple Z-stacks, with the 100x being a sub-stack of the 10x. (C) Zoom of B (top). Yellow arrow heads indicate mitotic cells, green arrow heads indicate interphase cells. (D) Zoom of B (bottom), showing two mitotic cells labeled with ch15 stripe paints and DAPI stain (top) or only stripe paints (middle). White and gray arrows indicate direction of spindle poles. Bottom, cartoon schematic, with the black outline representing the border of the DAPI stain. Black and gray arrows indicate direction of spindle poles.
(TIF)

**S10 Fig. Mature sperm in whole mount 5th instar larval testis.** Left: 10x image of whole mount larval testis stained with DAPI. White box indicates zoom shown to the right. Right: zoom of mature eupyrene sperm bundle labeled with DAPI.
(TIF)

**S11 Fig. Oligopaints in mature sperm from 5th instar larval testis squash.** Oligopaints labeling chromosomes 17 (red), 23 (cyan) and Z (yellow) in mature sperm from a 5th instar larval testis squash. DAPI is shown in gray.
(TIF)

**S12 Fig. CENP-T is present on meiotic chromosomes in larval testes.** (A) Schematic of intact metaphase I spindle and aligned chromosomes from larval testes and broken spindle with spread chromosomes after cytocentrifugation. (B) Widefield images from dissociated testes cytospreads showing representative cells (two) with CENP-T localization on meiotic chromosomes. Images show Z-projections of 5–10 Z stacks. Scale bars = 5 μm. (C) Left: Super-resolution image images showing CENP-T binding to a metaphase I cell. Right: cartoon schematic of a single chromosome (blue) with CENP-T kinetochores in green and microtubules (anti-Tubulin staining) in magenta.
(TIF)

**S13 Fig. Whole mount 5th instar larval ovary.** DAPI staining on whole mount 5th instar larval ovaries. Intact ovary is shown on top and below are loose ovarioles.
(TIF)

**S14 Fig. Super-resolution imaging of pachytene nuclei from 5th instar larval gonads.** (A) Super-resolution image of a representative field of cells in the zone of pairing (region 1) in whole mount larval ovaries labeled with ch7 stripe paints. (B) Super-resolution image of two representative pachytene cells in larval testes squashes (to maximize the possibility of observing space between the homologs) labeled with ch7 stripe paints.
(TIF)

**S1 Movie. Super-resolution image of mature 5th instar larval oocyte.** Animated Z-stack of super-resolution image of mature 5th instar larval oocyte (region 3b) stained with DAPI (gray).
(MOV)

**S1 File. Master data sheet.**
(XLSX)

## Acknowledgments

We would like to thank the members of the Lei and Drinnenberg labs, as well as Jamie Walters, Petr Nguyen, Martina Dalikova, Eric Joyce, and Son Nguyen for helpful discussion. We would also like to thank Catherine McManus, Jennifer Luppino, Stacie Hughes, Jean-René Huynh, and Anahi Molla-Herman for critical reading of the manuscript. Finally, we would like to thank Xufeng Wu and Jeff Reese in the NHLBI and NIDDK microscopy cores, respectively, for assistance with super-resolution microscopy.

## Author Contributions

**Conceptualization:** Leah F. Rosin.

**Data curation:** Leah F. Rosin, Jose Gil, Jr.

**Formal analysis:** Leah F. Rosin.

**Funding acquisition:** Leah F. Rosin, Ines A. Drinnenberg, Elissa P. Lei.

**Investigation:** Leah F. Rosin.

**Methodology:** Leah F. Rosin, Jose Gil, Jr.

**Project administration:** Elissa P. Lei.

**Resources:** Jose Gil, Jr.

**Supervision:** Ines A. Drinnenberg, Elissa P. Lei.

**Validation:** Leah F. Rosin.

**Visualization:** Leah F. Rosin.

**Writing – original draft:** Leah F. Rosin.

**Writing – review & editing:** Leah F. Rosin, Jose Gil, Jr., Ines A. Drinnenberg, Elissa P. Lei.

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
