## [Decision Letter · Decision Letter 0]

7 May 2021

Dear Dr Rosin,

Thank you very much for submitting your Research Article entitled 'Oligopaint DNA FISH as a tool for investigating meiotic chromosome dynamics in the silkworm, Bombyx mori' to PLOS Genetics.

The manuscript was fully evaluated at the editorial level and by four highly-qualified independent peer reviewers. Their recommendations ranged from ACCEPT (1), to MINOR REVISION (2), to MAJOR REVISION (1)  Obviously, this is an impressive and significant paper.  It seems very likely that suitably revised version will be accepted for publication. That said, we strong urge you to consider all of the comments, especially those of the most critical reviewer (#2).  WE share this reviewer's concern that:

"*Furthermore, almost all data is present without any quantitation. For example, how many oocytes or sperm are observed at each stage, and how many show the characteristics being reported? Some examples are below. By addressing these and the following comments, the authors may increase the accessibility and rigor of these results*."

This paper will provide a huge 're-start ' to the study of meiosis in Bombyx.  Providing the groundwork quantification will serve everyone who follows. One of us (RSH) also strongly echoes the this reviewer's suggestion that you avoid (entirely) the term 'elimination chromatin.  The term is very misleading. Moreover it easily gets confused with other processes such as chromatin dimunition.

We ask that you address the comments of other reviewers as well.  They are thoughtful and useful. We are convinced the revised manuscript will be impressive indeed.

(Note from RSH: Once again, I apologize for the near week long delay in issuing this decision letter. I have spent the last seven days coping with the flu - really, the flu?)

Now for the boilerplate:

We therefore ask you to modify the manuscript according to the review recommendations. Your revisions should address the specific points made by each reviewer.

[LINK]

Yours sincerely,

R. Scott Hawley

Associate Editor

PLOS Genetics

Wendy Bickmore

Section Editor: Epigenetics

PLOS Genetics

Reviewer's Responses to Questions

**Comments to the Authors:**

Reviewer #1: In Rosin et al. the authors use olipaints to look at chromosome pairing during female and male meiosis in the silkmoth Bombyx mori. Using the oligopaints to both full chromosomes and specific sections of the chromosomes showed that chromosomes undergo progressive pairing until fully paired along their lengths. In males the one end of the chromosome becomes unpaired while the other end remains associated when chromosomes become aligned. The authors suggest the unpaired end acts as a kinetochore to segregate the chromosomes at the first division.

The imaging beautifully shows the pairing of the chromosomes but the authors need to temper some of their claims before publication. The authors make several assumptions that need to be demonstrated in the manuscript or those claims need to be softened.

For example the authors claim in males the configuration they show when chromosomes are lined up are in metaphase I and the chromosomes are aligned towards the spindle poles but no images include a spindle. Therefore the data does not CLEARLY demonstrate that telomeric regions face poleward as stated in several places. The authors are assuming that a spindle is present and orients in that direction. Either spindle images with DNA at the same stage/ configuration need to be supplied or claim changed to “most likely” faces the spindle pole.

It is claimed that pairing is maintained after the loss of the central region on page 22 and after the SC has reformed in other locations. With no immunofluorescence images of the SC shown that are taken at the same stages as olipaints or taken with the oligopaints (the better experiment) the authors can only speculate about the status of the SC at the stages examined.

The authors need to be clearer in text, not just methods, the stages were classified based only on DAPI. Staging assessments were not collaborated using stage specific antibodies to the SC, determination markers, or stage specific structures as is standard in pairing studies in other meiotic systems and authors should be clear that the staging are estimated. This is particularily important if the authors did spread experiments, since in spreads temporal information is normally lost.

There is also confusion on whether the authors did chromosome squashes or spreads. The terms seem to be used interchangeably for the images in figure 2 but the techniques are different with different results. In spreads tissues and cells are disrupted and the free nuclei are spread on a slide, which is why temporal information is lost. In squashes a whole tissue is flattened onto a slide/ coverslip. While the tissue it distorted and nuclei can burst in the squashing process some temporal information is retained since cell burst in relation to their neighboring cells.

In the zone of pairing how does the author know which cells are the meiotic cells compared to somatic cells without a differentiation marker?

Minor notes

Line 86 should read segregation OF meiotic chromosomes

Line 239 needs a reference to who previously hypothesized the configuration in Bombyx.

Reviewer #2: This paper describes results the Authors have obtained applying oligo-paint technology to Bombyx mori meiosis. The results show that the meiotic chromosome organization and behavior can be studied using these techniques. The most interesting and significant results are in Figure 4 that show that one end of each bivalent ends can lead the chromosomes towards the poles. This is important because B. mori chromosomes lack localized centromeres in mitosis. Like C. elegans, these results show that one end of the chromosome assumes the role of a centromere by leading the way to the pole meiosis I pole. The fact that either end can assume this activity suggests this is like C. elegans, and a mechanism exists to restrict kinetochore-like activity to one end, like the position of the crossover. This makes the females extra interesting, because these chromosomes segregate without crossovers.

While these results are interesting, they mostly demonstrate the techniques and do not go into much depth. This is not to say these results are not important, only that they mostly represent a technological achievement. The paper could be improved with more rigor and quantitation, which would allow others to know what to expect when doing experiments in this system. A particularly valuable feature of both germlines is the array of meiotic stages that can be observed. However, it is difficult using the information in this paper to appreciate the organization of the germlines. Furthermore, almost all data is present without any quantitation. For example, how many oocytes or sperm are observed at each stage, and how many show the characteristics being reported? Some examples are below. By addressing these and the following comments, the authors may increase the accessibility and rigor of these results.

1) In Figures 2, 5 and 6, the organization of the germlines could be hard to understand. For example, in Figure 2F, why is diplotene between zygotene and pachytene. Schematics would the reader understand the relationship between Figure 2 and Figure 5A-C. Thus, for both the male and female germlines, a schematic showing the organization of the meiotic stages should be added. When showing meiotic metaphase or mitosis images, it would help to indicate the axis of the spindle (because there is no tubulin or spindle pole labels). Figure 6, line 342: how are the oocytes identified.

2) Line 89 and 386-388 – the idea the microtubules directly penetrate the chromatin should be stated with caution for two reasons. First, the lack of a CENP-A homolog, which is a centromere protein, does not mean there are no kinetochores. Second, kinetochore proteins are present on the chromosomes in meiosis of holocentric organisms like C. elegans, and they have an important role in meiotic chromosome segregation.

3) While “elimination chromatin” is used in prior publications, it is a confusing term and the authors should use it with some explanation. Is chromatin actually eliminated? What is it actually referring to.

4) Line 165: Describe in more detail how partially paired chromosomes are observed and define how often they are observed.

4) Figure S3 is important, and part of it could be a main figure with quantitation. This could replace some redundancy in existing regular figures. For example, some figures (eg. #2) show the same thing with different probes. Figures 3 and 4 show similar data, with 4 showing more stages and detail. Seeing the DAPI channel in 4A is not helpful and could be deleted and 4A,B,C show much the same thing.

5) Figure S5 – some images look like there is evidence of pairing in nurse cells. Please quantify.

6) Why is Figure S7 included or the text from lines 178-183. Explain the significance of this data or delete.

7) Lines 212-213 – this description is confusing because it is written that the chromosomes were divided into 5 or 13 stripes. But the rest of the discussion seems to concern the 5 stripes (first, middle, last) and not more reference to experiments with 13.

8) Line 238 “previously hypothesized” needs a reference.

9) Figure 4B: Could the authors could come up with more descriptive or useful terms for S1, S2, S3. Furthermore, the cartoon schematic is not accurate – it implies and X-form structure which is not obvious in the images.

10) Line 268-269: Explain how the whole mounts validate the findings. The images are not as clear. What is the significance of the apyrene nuclei? More description of how these images are important to the paper would help.

11) Figure 5: The authors conclude the leptotene homologs not paired. How is this defeind and measured and is it 100%? Some quantification is need because some nuclei seem to show some pairing. This is important because it indicates the state of the chromosomes before SC assembles. Conceivably this could be measured (eg the distance between each homolog).

12) Line 312 – define “polytrophic meroistic”

13) Lines 311-312 and 326: Is it surprising that chromosome-wide pairing persists in females? Do the cited references have data to suggest they should not be paired along their length? A review from von Wettstein (Ann Rev Genetics 1984) suggests the female meiotic chromosomes remain paired along their entire length by a modified SC structure. This analysis would also benefit from some quantitation to know if there is any variation. Because the chromosomes are small, this would help the conclusion that the chromosomes are paired.

14) Line 357: In most organisms that SC components are poorly conserved. In mammals there are SC components with no known paralogs in other organisms. Therefore, Drosophila is not that unusual, and indeed C(3)G is a ortholog of SC proteins in other organisms.

Reviewer #3: In this manuscript, Rosin et al. employ Oligopaint DNA FISH to examine homolog pairing and compaction during both male and female meiosis in the holocentric Bombyx mori. This study shows Oligopaints can be used in both squashed and whole tissue with high specificity in this silkworm moth. It also reveals that telomeric regions face poleward, likely acting as localized kinetochores, with either telomere having an equal probability of facing poleward and potentially explaining how these chromosomes segregate in the absence of CENP-A. This is a very interesting study that shows the use of Oligopaints for the first time to assess compaction and pairing of multiple, whole chromosomes in both male and female meiosis and is the first study of this kind in Lepidoptera, opening up various avenues for future discoveries about meiosis in B. mori. This is a well-written manuscript with beautiful images and striking observations that showcases the importance of looking at meiosis in different species and will be of interest to the broad readership of PLoS Genetics. Below are some questions and recommended revisions:

Major:

1- Page 6, Line 117: Please add a few words in the main text explaining the rationale behind the choice of the six autosomes and the Z sex chromosome being targeted in this analysis (i.e. were the six autosomes selected at random or for specific reasons from among the 27 autosomes in this organism? The rationale for why the W chromosome is not targeted is mentioned in the legend for Fig S5, but it would be helpful to raise this point up front in the main text when describing why the Z is targeted).

2- Page 8: The authors report noticing that all chromosomes do not pair simultaneously during early meiotic prophase I in larval testes squashes. Are there specific chromosomes among those assessed that are achieving pairing always earlier or later? Is there something inherently unique about these if that is the case (i.e. chromosome size, enrichment for germline-expressed genes, heterochromatin/euchromatin distribution, faster/slower replication prior to meiotic entry…)?

3- The authors should indicate the frequency with which they observed the large chromosome loops (Fig 3A) and what they think they represent? Are these indicating that pairing does not start at a single region or chromosome end like in worms, but instead at multiple regions between the homologs (or at opposite ends of the chromosomes?)? Can the authors correlate this with either how SC assembly unfolds in this organism (i.e. progressively from one end or from multiple nucleation sites throughout the chromosomes) or location of double-strand break sites?

4- Please move Figure S8 into main Fig 5 since this will be very helpful for those not familiar with staging in the testicular lobe in Bombyx mori. This will also be consistent with what was done in Fig 6 by including a useful schematic of the 5th instar larval ovary in panel A.

5- Fig 5 C and D- There seems to be a comet tail signal revealed with S2 and S3 on metaphase I chromosomes. Can the authors briefly explain this either in the main text or figure legend?

Minor:

1- Reference #40 is duplicated as #45.

2- Page 4, Line 86- Typo (word missing) in “for segregation meiotic chromosomes…”

3- Page 6, Line 123- Reference is cited instead of reference #.

4- Fig S2: Please use colored arrows or arrows and arrowheads to indicate the mitotic and meiotic clusters.

5- Figure 2 - legend indicates “D, larval ovary”, but panel D is labeled as Testes in the figure. Also, correct where it says “metaphase I (F) cells” to (G) instead?

6- Please indicate number of larvae examined in the legend for each figure.

7- Figure legend for Fig S7- Typo “in the this cell line”.

Reviewer #4: In their manuscript, Rosin and colleagues analyzed the pairing of maternal and paternal chromosomes during the first steps of meiosis in Bombyx mori. They used thousands of fluorescent oligos to label whole chromosomes or specific subregions of single chromosomes. With these tools, they analyzed chromosome behavior at each step of meiotic prophase I on fixed germ cells in male and female silkworm.

The images are stunning. I don’t have any major criticism. Obviously, this study is descriptive by nature, but it sets the stage for future functional studies. It will be a reference in this field.

My comments are rather curiosity questions:

1) Do chromosome start to pair by the same regions?

2) Is there an order in chromosome pairing?

3) When are double-strand breaks formed compared to synapsis? Is it possible to perform double-labelling with other meiotic landmarks? Such as the synaptonemal complex or DSBs?

Minor comments:

1) I would add more introductory material. Especially, regarding the C. elegans literature, which may not be obvious to every reader.

**Have all data underlying the figures and results presented in the manuscript been provided?**

Reviewer #1: Yes

Reviewer #2: Yes

Reviewer #3: Yes

Reviewer #4: Yes

PLOS authors have the option to publish the peer review history of their article (what does this mean?). If published, this will include your full peer review and any attached files.

Reviewer #1: No

Reviewer #2: No

Reviewer #3: No

Reviewer #4: No

---

## [Editor Report · Decision Letter 1]

7 Jul 2021

Dear Dr Rosin,

We are pleased to inform you that your manuscript entitled "Oligopaint DNA FISH reveals telomere-based meiotic pairing dynamics in the silkworm, Bombyx mori" has been editorially accepted for publication in PLOS Genetics. Congratulations! We thank you enormously for careful response to the reviewers'  concerns and comments.  We feel that you have more than fully addressed the concerns about quantification and terminology.  We are aware that you were fine with ending out for re-review - but see no reason to do so. This is a superb paper that will revitalized a field and a model organism and popularize the technologies you have developed. 

Yours sincerely,

R. Scott Hawley

Associate Editor

PLOS Genetics

Wendy Bickmore

Section Editor: Epigenetics

PLOS Genetics

Comments from the reviewers (if applicable):

**Data Deposition**

http://datadryad.org/submit?journalID=pgenetics&manu=PGENETICS-D-21-00432R1

**Press Queries**

---

## [Editor Report · Acceptance letter]

23 Jul 2021

PGENETICS-D-21-00432R1 

Oligopaint DNA FISH reveals telomere-based meiotic pairing dynamics in the silkworm, Bombyx mori 

Dear Dr Rosin, 

We are pleased to inform you that your manuscript entitled "Oligopaint DNA FISH reveals telomere-based meiotic pairing dynamics in the silkworm, Bombyx mori" has been formally accepted for publication in PLOS Genetics! Your manuscript is now with our production department and you will be notified of the publication date in due course.

With kind regards,

Katalin Szabo

PLOS Genetics

On behalf of:
